# Patritumab deruxtecan in HER2-negative breast cancer: part B results of the window-of-opportunity SOLTI-1805 TOT-HER3 trial and biological determinants of early response

Patritumab deruxtecan (HER3-DXd) exhibits promising efficacy in breast cancer, with its activity not directly correlated to baseline *ERBB3*/HER3 levels. This research investigates the genetic factors affecting HER3-DXd's response in women with early-stage hormone receptor-positive and HER2-negative (HR +/HER2-) breast cancer. In the SOLTI-1805 TOT-HER3 trial, a single HER3-DXd dose was administered to 98 patients across two parts: 78 patients received 6.4 mg/kg (Part A), and 44 received a lower 5.6 mg/kg dose (Part B). The CelTIL score, measuring tumor cellularity and infiltrating lymphocytes from baseline to day 21, was used to assess drug activity. Part A demonstrated increased CelTIL score after one dose of HER3-DXd. Here we report CelTIL score and safety for Part B. In addition, the exploratory analyses of part A involve a comprehensive study of gene expression, somatic mutations, copy-number segments, and DNA-based subtypes, while Part B focuses on validating gene expression. RNA analyses show significant correlations between CelTIL responses, high proliferation genes (e.g., *CCNE1, MKI67*), and low expression of luminal genes (e.g., *NAT1, SLC39A6*). DNA findings indicate that CelTIL response is significantly associated with *TP53* mutations, proliferation, non-luminal signatures, and a distinct DNA-based subtype (DNADX cluster-3). Critically, low HER2DX *ERBB2* mRNA, correlates with increased HER3-DXd activity, which is validated through in vivo patient-derived xenograft models. This study proposes chemosensitivity determinants, DNA-based subtype classification, and low *ERBB2* expression as potential markers for HER3-DXd activity in HER2-negative breast cancer.

HER3 plays a crucial role in oncogenic signaling and is a compelling therapeutic target[1–3]. However, clinical development of antibodies against HER3 has been unsuccessful due to suboptimal response rates in clinical trials across tumor types. Patritumab deruxtecan (HER3-DXd) is a first-in-class HER3-directed antibody-drug conjugate (ADC) composed of a fully human anti-HER3 IgG1 monoclonal antibody covalently linked to an exatecan derivative topoisomerase I inhibitor payload via a tetrapeptide-based cleavable linker[4]. HER3-DXd was granted a breakthrough therapy designation by the Food and Drug Administration (FDA) in 2021 for advanced *EGFR*-mutated non-small cell lung cancer[5].

✉e-mail: alprat@clinic.cat

In breast cancer, HER3-DXd has shown promising activity across all subtypes, including hormone receptor-positive/HER2-negative (HR+/HER2-) disease. In a phase 1-2 clinical trial, HER3-DXd monotherapy was tested in 113 patients with heavily pre-treated HER3-expressing HR+/HER2- metastatic breast cancer. The overall response rate (ORR) and median progression-free survival (PFS) were 30.1% and 7.4 months, respectively[6]. In the SOLTI-1805 TOT-HER3 window-of-opportunity trial Part A, a single dose of HER3-DXd (6.4 mg/kg) was evaluated in 78 patients with untreated early-stage HR+/HER2- breast cancer[7]. In Part B, a lower single dose of HER3-DXd (5.6 mg/kg) was evaluated in 20 patients with untreated early-stage HR+/HER2- breast cancer and 17 patients with triple-negative breast cancer (TNBC)[8]. The primary endpoint of SOLTI-1805 TOT-HER3 of the trial was to evaluate changes in the CelTIL score.

Despite the encouraging activity of HER3-DXd in HR+/HER2- breast cancer, the biological determinants of its efficacy are vastly unknown. In fact, no relationship has been observed between HER3-DXd efficacy and the expression of HER3 protein by immunohistochemistry (IHC), or *ERBB3* expression by mRNA[6]. In line with this, trastuzumab deruxtecan (T-DXd), another ADC that contains the same linker and payload as HER3-DXd, has demonstrated clinically meaningful activity across all levels of HER2 expression, with the highest efficacy in HER2-positive tumors[9]. Thus, the identification of biomarkers that predict response to each DXd-based ADC might help guide the clinical development of both drugs. Overall, there is an urgent need to identify biomarkers of response to ADCs in solid tumors, and simple target levels do not seem to be sufficient[10]. The window of opportunity SOLTI-1805 TOT-HER3 trial offered the opportunity for exploration of genomic biomarkers and to gain insights into which patients are likely to benefit from HER3-DXd. This trial and biomarker approach could be applicable to any ADC and other cancer types.

Here, we report the results of SOLTI-1805 TOT-HER3 part B trial and the results of the exploratory study, including genomic analyses on baseline tumor samples from the SOLTI-1805 TOT-HER3 trial, with the aim of better understanding the biological features associated with early response from HER3-DXd monotherapy. We show an increased CelTIL score after one dose of HER3-DXd in part B, and reveal gene expression and DNA-based determinants of early response to HER3-DXd.

## Results

### CelTIL score and clinical response in early-stage HER2-negative breast cancer

In SOLTI-1805 TOT-HER3 part A, 78 patients with HR+/HER2- breast cancer received a single dose of 6.4 mg/kg HER3-DXd (Fig. 1a). In part B, a total of 44 patients were assessed for eligibility (Supplementary Fig. 1a), and 20 patients with HR+/HER2- breast cancer and 17 patients with TNBC were enrolled in the study to receive a single dose of 5.6 mg/kg HER3-DXd (Fig. 1a). A mandatory tumor biopsy was performed at C1D21. The primary objective of the trial was to evaluate changes in the CelTIL score, which is a combined biomarker that integrates the proportion of tumor cellularity and TILs on the same biopsy[11–13]. High CelTIL scores identify tumors that are highly immune infiltrated with reduced tumor cellularity. CelTIL is an early indicator of drug activity, and potentially an early indicator of long-term drug efficacy. Part A demonstrated an increase CelTIL score after one dose of HER3-DXd, and the main results were reported elsewhere[7].

In part B, reported here for the first time, the mean age of the study population ($n = 37$) was 51 years (range, 30–81 years); 20 patients (54%) were premenopausal, and most patients presented with histological grade 2 (30%) or 3 tumors (51%) and invasive ductal carcinoma (89%). The median tumor size was 21 mm, with a range of 10-80 mm. Most patients had *ERBB3*-ultralow tumors (57%). HER3 protein expression was evaluated in 24 patients (65%) and most demonstrated

high overall HER3 membrane expression (79%). PAM50 subtype distribution was 46% Basal-like, 24% Luminal A, 19% Luminal B, 11% HER2-enriched (Supplementary Table 1). The primary objective of part B was the increase in CelTIL score after one dose of HER3-DXd. Overall, a mean increase in CelTIL score of 9.4 ($p = 0.046$) was detected (Supplementary Fig. 1b); with mean differences of 2.2 and 17.9 in HR+/HER2- and TNBC, respectively. Of the 37 patients, 34 were assessable for clinical response. The clinical ORR was 32% (30% in HR+/HER2− and 35% in TNBC). A significant increase in CelTIL score (mean increase = 23.9, $p = 0.043$) was detected in patients with clinical response (Supplementary Fig. 1b). Secondary objectives of part B included the association of *ERBB3* mRNA with CelTIL at day 21 and safety. *ERBB3* mRNA expression was not associated with CelTIL variation (Supplementary Fig. 1c). Additionally, safety in part B was similar to that observed in part A (Supplementary Table 2), with lower incidence of hematological and hepatic toxicity compared to Part A. No interstitial lung disease nor grade 4/5 events were observed (Supplementary Table 3).

To provide further evidence of the association of C1D21 CelTIL with drug efficacy, we combined SOLTI-1805 TOT-HER3 parts A and B, and observed that a consistent CelTIL increase was significantly (p<0.001) associated with clinical response (Fig. 1c). Additionally, in SOLTI-1007 NeoEribulin trial[14], an increase of CelTIL after 1 cycle of eribulin monotherapy (i.e., at cycle 2 day 1) was significantly associated with residual cancer burden 0 or 1 (RCB-0/1) at surgery after the completion of 4 cycles in patients with HER2-negative breast cancer (Supplementary Fig. 2a, b). Overall, these results suggest that CelTIL in HER2-negative breast cancer is an early readout of drug activity, and potentially an early indicator of drug efficacy and long-term benefit.

### SOLTI-1805 TOT-HER3 translational study design

In the exploratory analyses of SOLTI-1805 TOT-HER3, RNA was isolated from 77 baseline pre-treatment formalin-fixed paraffin-embedded (FFPE) from part A of SOLTI-1805 TOT-HER3, and from 20 FFPE tumor samples from part B for validation purposes. DNA was obtained from 49 baseline pre-treatment FFPE tumor samples from part A. Patient characteristics are summarized in Supplementary Table 4. To identify biomarkers of early response to HER3-DXd, we evaluated the association between each baseline variable and CelTIL response (Fig. 1), defined as an absolute increase of CelTIL of ≥20 points between the two time points (i.e., C1D21 minus baseline). In part A, an absolute increase of CelTIL of ≥20 points predicted clinical response with an AUC of 0.68 ($p = 0.009$) (Supplementary Fig. 3).

### Correlative analysis of CelTIL response and mRNA expression

After a single dose of HER3-DXd, 51 (66%) and 26 (34%) tumors had a low and high CelTIL response, respectively. At baseline, the expression of 41 of 185 (22.2%) genes was found associated with CelTIL response using SAM analysis with FDR<10% (Fig. 2a). Of the 41 genes, 37 (90.2%) were also found associated with CelTIL response using univariate logistic regressions. Overall, high expression of proliferation and cell division-related genes such as *AURKA* ($p = 0.002$), *CCNE1* ($p = 0.014$), and *MKI67* ($p = 0.010$) was associated with CelTIL response. Conversely, high expression of luminal-related genes such as *NAT1* ($p = 0.019$), *SLC39A6* ($p = 0.033$), *MAGED2* ($p = 0.021$) and *THSD4* ($p = 0.029$) was associated with a lack of CelTIL response (Fig. 2b).

At baseline, 9 of 12 (75.0%) gene expression signatures were found associated with CelTIL response. High scores of the PAM50 Basal-like ($p = 0.013$), PAM50 HER2-enriched ($p = 0.044$), PAM50 Proliferation ($p = 0.002$), PAM50 risk of recurrence (ROR) ($p = 0.002$) and HER2DX Proliferation ($p = 0.031$) signatures were associated with CelTIL response. High scores of the PAM50 Luminal A ($p = 0.001$), PAM50 Normal-like ($p = 0.036$), PAM50 chemo-endocrine sensitive (CES) ($p =$

0.003) and HER2DX HER2 amplicon ($p$ = 0.005) signatures were associated with lack of CelTIL response (Supplementary Fig. 4). Of note, even though the Luminal A and the Basal-like signatures were associated with CelTIL response, estrogen receptor (ER) and progesterone receptor (PR) protein expression were not (Supplementary Fig. 5).

### ERBB2/HER2 expression and HER3-DXd response

Our pre-treatment RNA-based analysis identified *ERBB2* mRNA and HER2 amplicon signature as determinants of early response to HER3-DXd.

Specifically, low expression of HER2DX *ERBB2* and HER2DX HER2 amplicon signature was found statistically significantly associated with CelTIL response as a continuous variable ($p$ = 0.006 and $p$ = 0.005, respectively) (Fig. 3a). We also defined HER2DX *ERBB2* and HER2DX HER2 amplicon signature group categories based on the tertiles of the TOT-HER3 part A (Supplementary Fig. 6). HER2DX *ERBB2* and HER2DX HER2 amplicon signature group categories defined by tertiles were found statistically significantly associated with lack of CelTIL response ($p$ = 0.019 and p<0.001, respectively), with the proportion of patients with a CelTIL response in HER2DX *ERBB2* mRNA low-tertile,

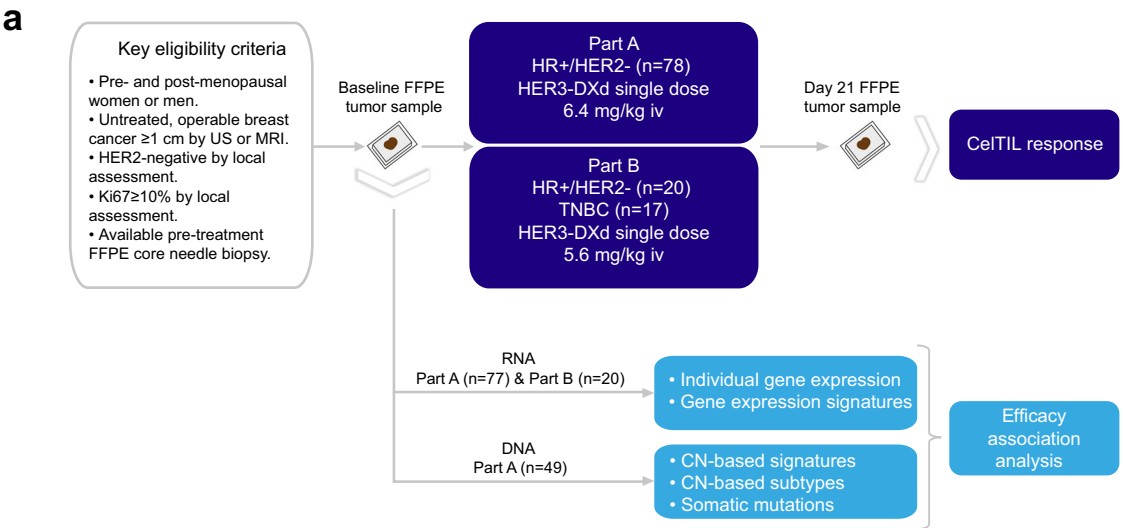

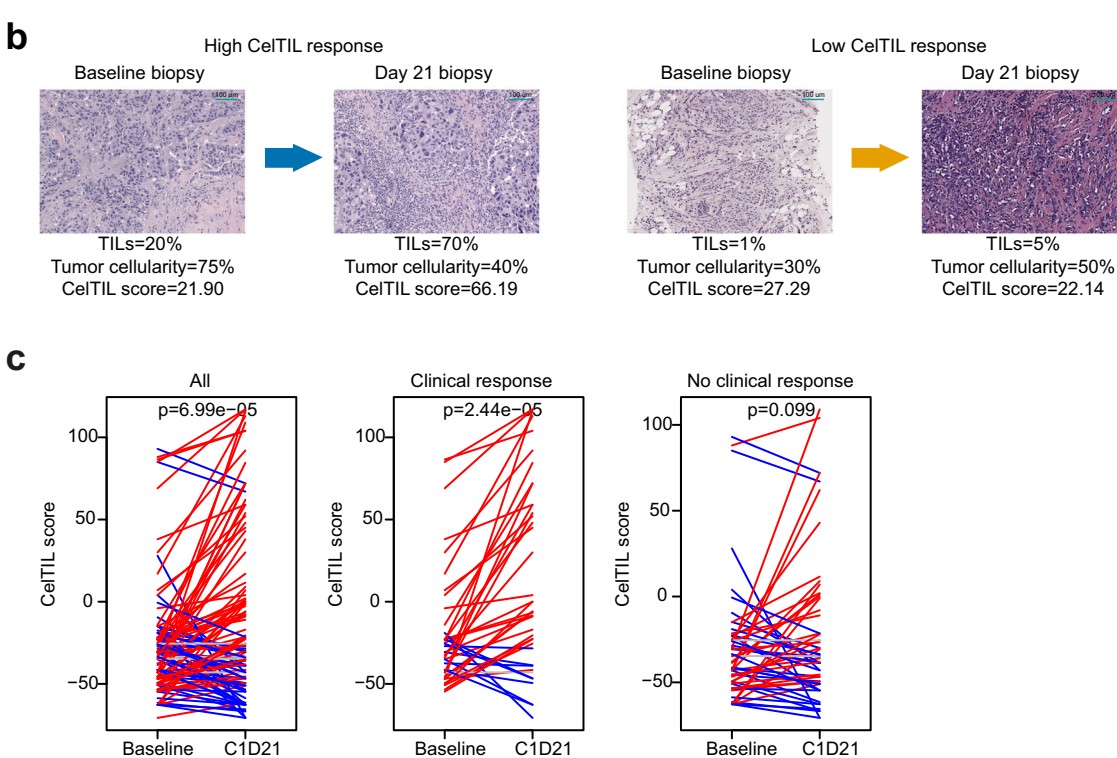

**Fig. 1 | SOLTI-1805 TOT-HER3 clinical trial and associated translational study framework. a** SOLTI-1805 TOT-HER3 trial design and translational study design. **b** Representative examples of H&E staining of a patient's tumor with high CelTIL response and a patient's tumor with low CelTIL response. H&E stainings were conducted for all baseline ($n$ = 114) and day 21 samples ($n$ = 114). **c** CelTIL score change in 114 samples of the SOLTI-1805 TOT-HER3 trial part A and part B combined in all patients, in patients with clinical response at day 21 ($n$ = 40), and in patients without clinical response at day 21 (C1D21) ($n$ = 56). Red lines represent increases at day 21 while blue lines represent decreases at day 21. *P*-values (p) were determined by two-tailed unpaired $t$ tests. Source data are provided as a Source Data file.

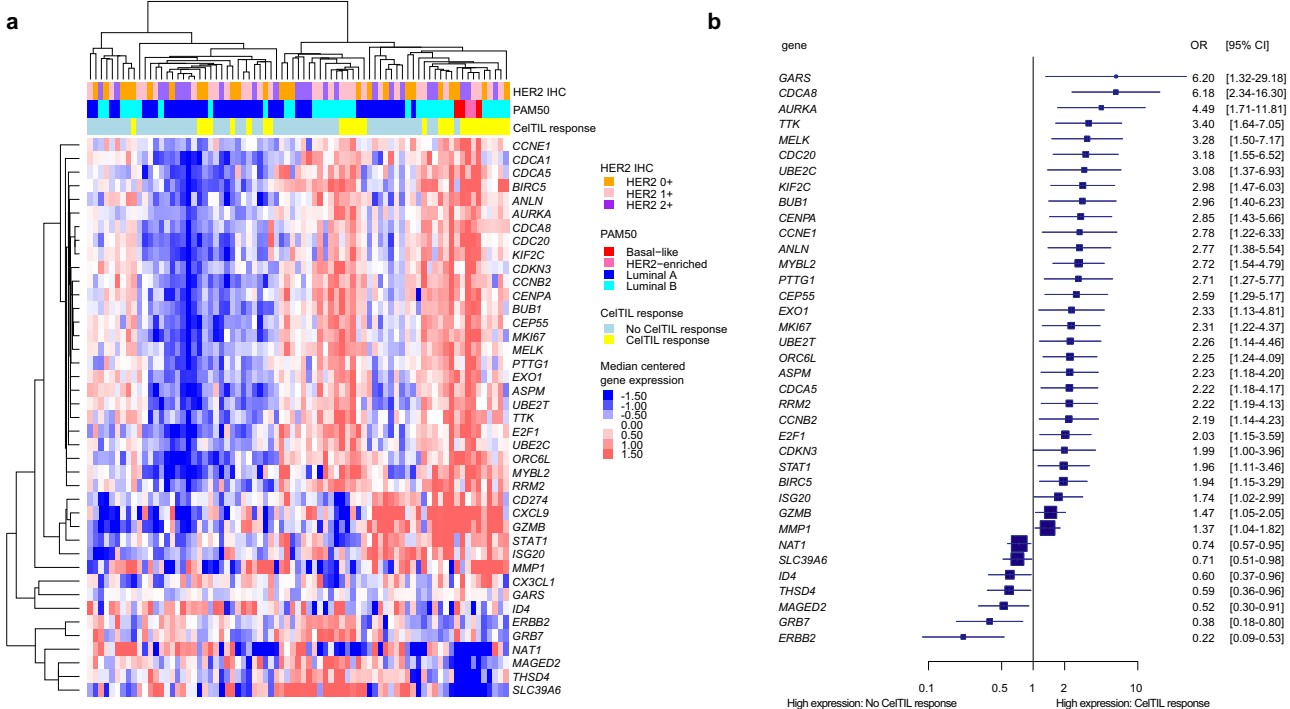

**Fig. 2 | Association of baseline gene expression with CelTIL response after one dose of HER3-DXd. a** Unsupervised hierarchical clustering of 41 genes were associated with CelTIL response after one dose of HER3-DXd in an unpaired SAM analysis (FDR<10%). **b** Forest plot of 37 of genes significantly associated with CelTIL

response after one dose of HER3-DXd in a logistic regression analysis in patients with CelTIL response (*n* = 26) and without CelTIL response (*n* = 51). Data are presented as the odds ratios (OR) with error bars showing 95% confidence intervals. Source data are provided as a Source Data file.

medium-tertile and high-tertile groups being 60.0%, 26.3% and 21.1%, respectively (Fig. 3b). The distribution of HER2 IHC status in the 77 baseline tumor samples analyzed was 32% HER2 0+, 38% HER2 1+ and 30% HER2 2+ (Fig. 3c).

Similar to HER2DX *ERBB2* expression, HER2 protein levels by IHC were found inversely associated with CelTIL response (HER2 0 vs HER2 2+, *p* = 0.043) and HER2 2+ had a lower proportion of patients with CelTIL response (13.0%) compared to HER2 0 (40.0%) and 1+ (45.0%) (*p* = 0.034) (Fig. 3d). As expected, HER2DX *ERBB2* mRNA was found differentially expressed across HER2 0, 1+ and 2+ cases (Fig. 3e, f). HER2DX *ERBB2* mRNA was not associated with any PAM50 subtype in part A (Supplementary Fig. 7a). HER2DX *ERBB2* mRNA was moderately correlated with *ERBB3* mRNA but no significant correlation was observed between HER2DX *ERBB2* mRNA or HER2 protein levels by IHC and HER3 protein levels by IHC (Supplementary Fig. 7b, c). A sensitivity analysis using different cutoffs of CelTIL response (>0, ≥10, ≥20, ≥30, ≥40) was performed. HER2DX *ERBB2* mRNA levels were inversely associated with CelTIL response in all explored cutoffs, with the highest association at CelTIL response cutoff ≥20 (Supplementary Fig. 8).

Next, we evaluated the ability of HER2DX *ERBB2* expression to predict CelTIL response independently of other molecular variables. Across various logistic regression bivariate analyses, higher HER2DX *ERBB2* mRNA expression was significantly associated with lower CelTIL response after adjusting for HER2 IHC (*p* = 0.002), PAM50 subtype (*p* = 0.001), PAM50 luminal vs non-luminal (*p* = 0.001), PAM50 proliferation (*p* = 0.003) and PAM50 ROR (*p* = 0.003) (Fig. 3g). Interestingly, the association of HER2 IHC with CelTIL response was lost (*p* = 0.538) when HER2DX *ERBB2* expression was included in the logistic regression model.

Finally, we tested the ability of HER2DX *ERBB2* mRNA expression to predict CelTIL response in an independent dataset of 20 patients with early-stage HR+/HER2− breast cancer treated in Part B of the trial.

HER2DX *ERBB2* expression as a continuous variable was found statistically significantly associated with no CelTIL response (AUC = 0.783) (Supplementary Fig. 9a). According to the HER2DX *ERBB2* tertiles identified in part A, the proportion of patients with a CelTIL response across HER2DX *ERBB2* low-tertile, medium-tertile and high-tertile groups identified in part B was 40%, 29%, and 0%, respectively (Supplementary Fig. 9b).

We next sought to understand the association between *ERBB2* expression and response to chemotherapy in two cohorts of patients with HER2-negative breast cancer (i.e., a neoadjuvant taxane-anthracycline-based cohort, hereafter GSE25066[15], and the neoadjuvant eribulin SOLTI-1007-NeoEribulin cohort[14]). *ERBB2* mRNA was not associated with response (i.e., pCR or CelTIL response) to chemotherapy (Supplementary Fig. 10).

Finally, as an additional validation of *ERBB2* as a potential mechanism of resistance to HER3-DXd, we assessed HER2DX *ERBB2* in the baseline tumors of a cohort of 30 breast cancer patient-derived xenografts (PDXs) treated with HER3-DXd[16]. Ninety percent of PDXs had experienced a complete response or partial response but 53% had a relapse[16]. PDXs were classified as relapsed vs non-relapsed. Although the measurement of resistance was very different to that of the SOLTI-1805 TOT-HER3 study, HER2DX *ERBB2* was significantly higher in baseline tumors of PDXs that relapsed compared to those that did not relapse, further indicating towards a role of HER2DX *ERBB2* in the response of this ADC (Supplementary Fig. 11).

## DNA somatic mutations and CelTIL response

Forty-nine of 77 patients (64%) in SOLTI-1805 TOT-HER3 part A had tumor DNA-sequencing data available. Baseline characteristics of the 49 patients are summarized in Supplementary Table 1. CelTIL changes between the RNA (*n* = 77) and DNA (*n* = 49) cohorts were comparable. All patients had at least 1 gene somatically mutated. The most frequently mutated genes were *PIK3CA* (33%), *GATA3* (29%), *ATM* (24%),

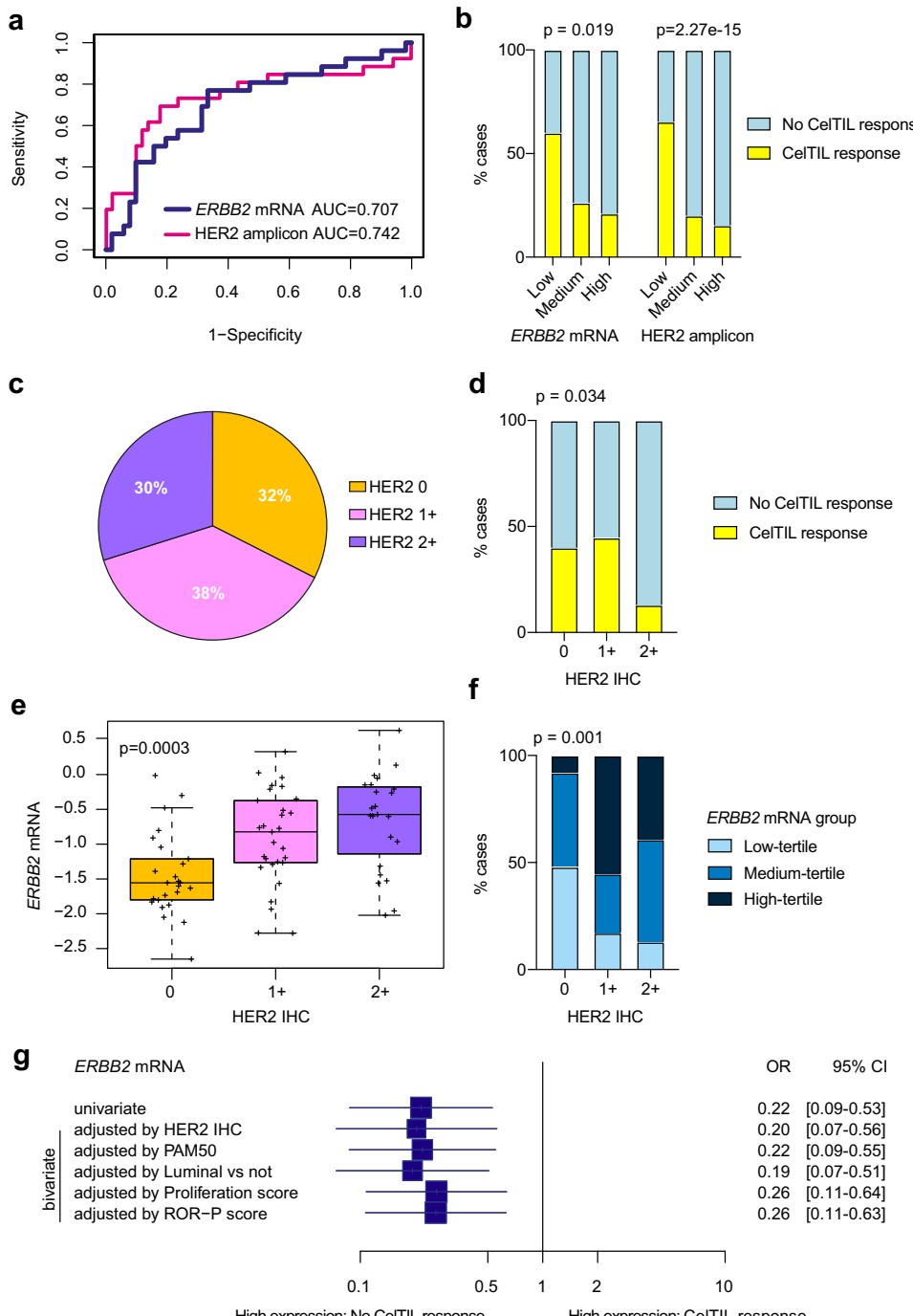

**Fig. 3 | High baseline HER2DX *ERBB2* mRNA is associated with low CelTIL response after one dose of HER3-DXd. a** Performance of the HER2DX *ERBB2* mRNA and HER2 amplicon signature score to predict CelTIL response after one dose of HER3-DXd. ROC AUC values are reported. **b** Proportion of tumors with high and low CelTIL response after one dose of HER3-DXd in each HER2DX *ERBB2* group (as defined by tertiles: low *n* = 26, medium *n* = 26, high *n* = 25) and each HER2 amplicon signature group (as defined by tertiles low *n* = 26, medium *n* = 26, high n = 25). *P*-values (p) were determined by two-sided Fisher's exact test. **c** Distribution of HER2 IHC status (HER2 0 *n* = 25, HER2 1+ *n* = 29, HER2 2+ *n* = 23) in the 77 baseline samples analyzed. **d** Proportion of tumors with high and low CelTIL response after one dose of HER3-DXd in each HER2 IHC group (HER2 0 *n* = 25, HER2 1+ *n* = 29, HER2 2+ n = 23). *P*-value (p) was determined by two-sided Fisher's exact test.

**e** Boxplot showing HER2DX *ERBB2* mRNA expression across HER2 IHC groups (HER2 0 *n* = 25, HER2 1+ *n* = 29, HER2 2+ *n* = 23). For the boxplot, center line indicates median; box limits indicate upper and lower quartiles; whiskers indicate 1.5× interquartile range. *P*-value (p) was determined by one-way ANOVA. **f** Proportion of tumors with high, medium and low HER2DX *ERBB2* mRNA (as defined by tertiles: low *n* = 26, medium *n* = 26, high *n* = 25) in each HER2 IHC group. *P*-value (p) was determined by two-sided Fisher's exact test. **g** Forest plot of the association of HER2DX *ERBB2* mRNA with CelTIL response in univariate and bivariate logistic regression analyses in patients with CelTIL response (*n* = 26) and without CelTIL response (*n* = 51). Data are presented as the odds ratios (OR) with error bars showing 95% confidence intervals. Source data are provided as a Source Data file.

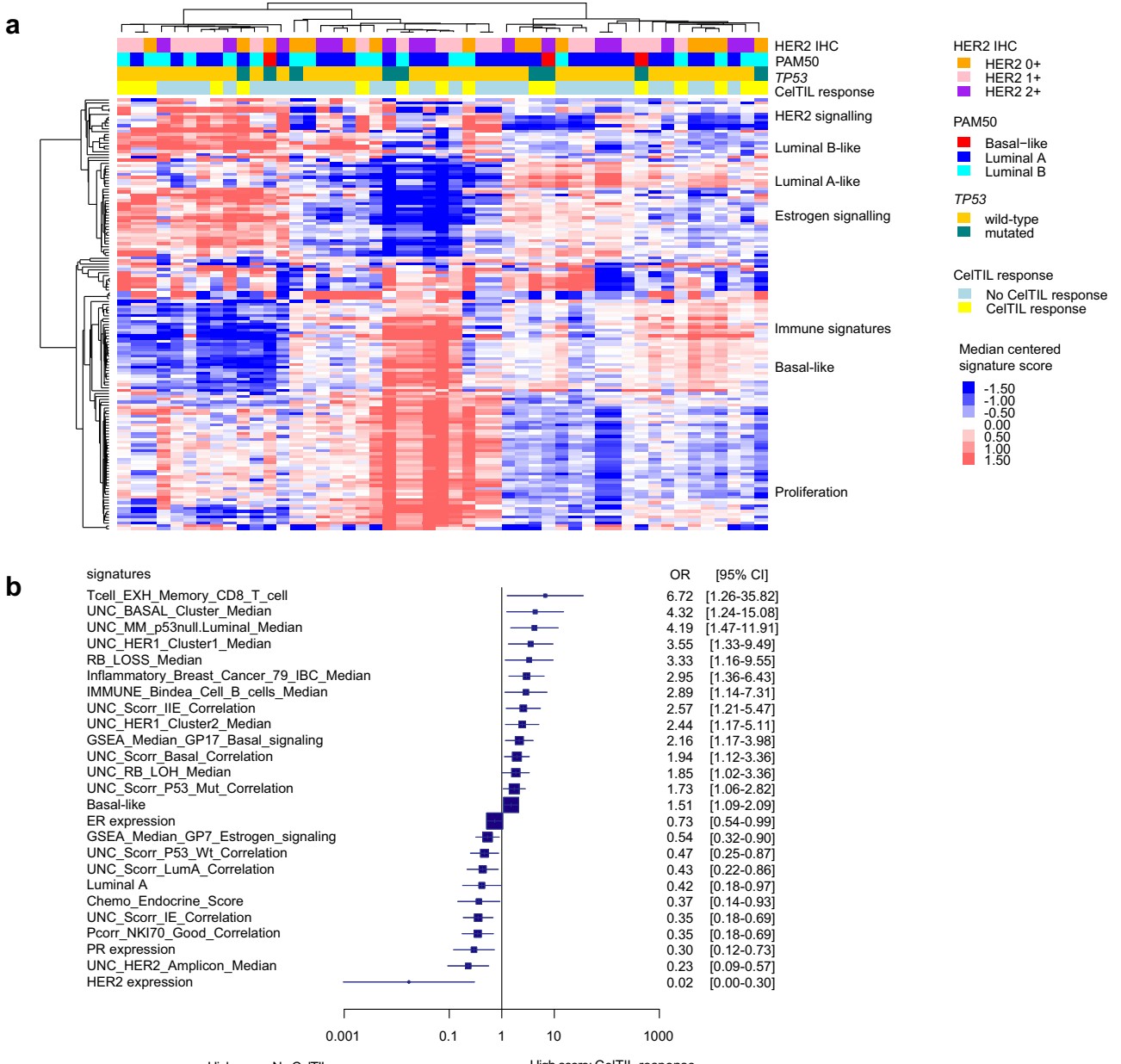

**Fig. 4 | Association of DNADX CN-based signatures associated with CelTIL response after one dose of HER3-DXd. a** Unsupervised hierarchical clustering of 150 CN-based signatures associated with CelTIL response after one dose of HER3-DXd in an unpaired SAM analysis (FDR<10%). **b** Forest plot of a selection of CN-based signatures significantly associated with CelTIL response after one dose of HER3-DXd in a logistic regression analysis in patients with CelTIL response (*n* = 16) and without CelTIL response (*n* = 33). Data are presented as the odds ratios (OR) with error bars showing 95% confidence intervals. Source data are provided as a Source Data file.

*CDH1* (20%), *TP53* (18%), *KMT2C* (16%), *KMT2D* (16%), and *MAP3K1* (14%). The frequency of *TP53* mutations was numerically higher (i.e., 100.0% vs. 10.7%) in Basal-like tumors (*n* = 3/3) compared to Luminal A tumors (*n* = 3/28) (Supplementary Fig. 12a). Among the most frequently mutated genes, CelTIL response rate was higher in *TP53* mutated tumors compared to *TP53* wild-type tumors (66.7% vs. 25.0%; Fisher's exact test *p* = 0.043) (Supplementary Fig. 12b). In logistic regression analyses, *TP53* mutations were significantly associated with CelTIL response in univariate analysis (odds ratio = 6.00, *p* = 0.024) but this significance was lost after adjusting by PAM50 subtype (*p* = 0.068), proliferation (*p* = 0.103) and ROR scores (*p* = 0.110) probably due to the small sample size. Of note, 4 of 49 tumors (8.2%) had mutations in *ERBB2*, and this alteration was not found associated with CelTIL response (*p* = 0.993).

## DNA copy-number (CN)-based signatures and CelTIL response

Next, we identified 150 DNA CN-based signatures previously defined to capture RNA- and protein-based phenotypes such as the PAM50-related biology[17,18]. A total of 90 of 150 (60.0%) DNADX signatures were significantly associated with CelTIL response (FDR<10% and univariate logistic regression) (Fig. 4a and Supplementary Fig. 13). Concordant with the gene expression data, high scores of signatures related to the Basal-like biology (e.g., UNC_BASAL_Cluster_Median) and high proliferation (e.g., UNC_Scorr_IIE_Correlation and UNC_RB_LOH_Median) were associated with CelTIL response, while high scores of signatures related to the Luminal A biology (e.g., UNC_Scorr_IE_Correlation and UNC_Scorr_LumA_Correlation) and HER2 (e.g., HER2 expression and UNC_HER2_Amplicon_Median) were associated with lack of CelTIL response (Fig. 4b).

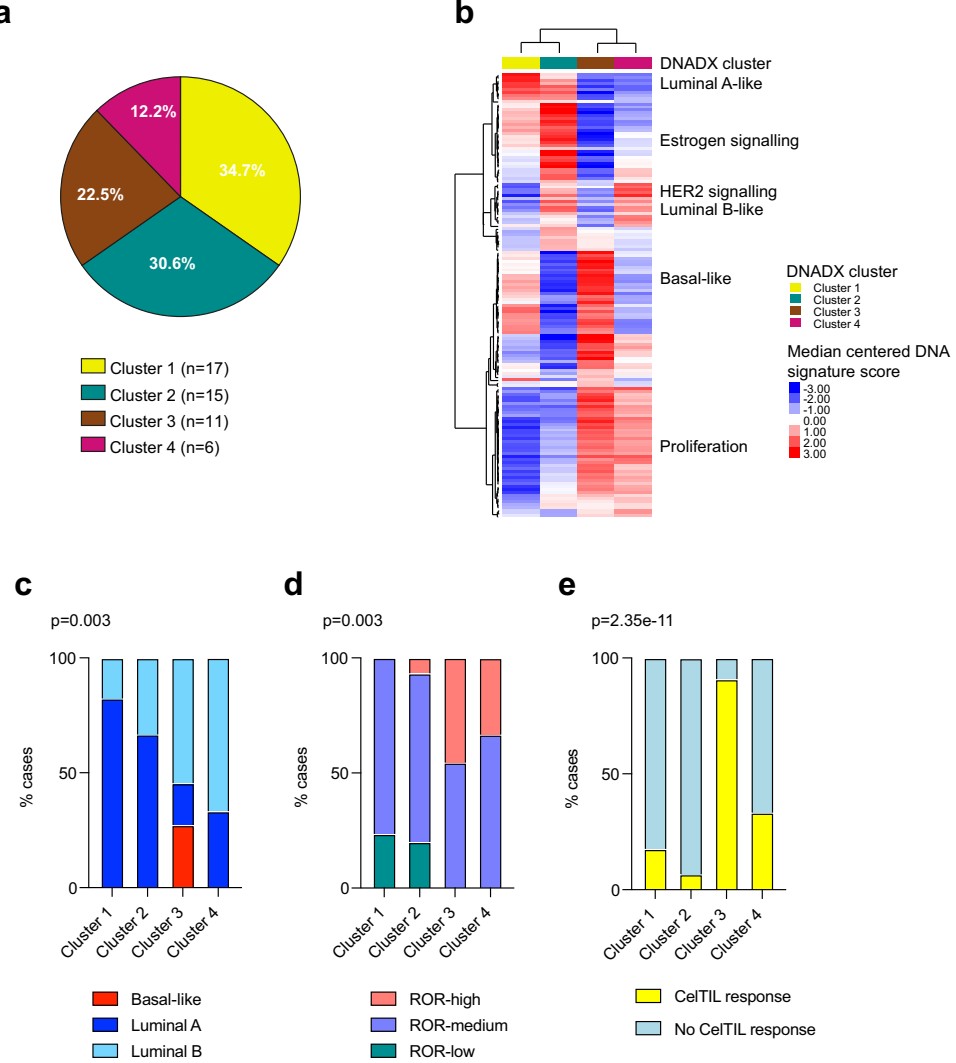

**Fig. 5 | Characteristics of the DNADX CN-based subtypes in the TOT-HER3 population with tumor NGS DNAseq data. a** Distribution of DNADX CN-based subtypes in 49 tumors of the TOT-HER3 trial (Cluster 1 *n* = 17, Cluster 2 *n* = 15, Cluster 3 *n* = 11, Cluster 4 *n* = 6). **b** Unsupervised hierarchical clustering of 150 CN-based signatures in each CN-based subtype. The correlation (Cor) values of signatures tracking similar phenotypes are reported near the dendogram. **c** Proportion of PAM50 molecular subtypes in each DNADX CN-based subtype (Cluster 1 *n* = 17, Cluster 2 *n* = 15, Cluster 3 *n* = 11, Cluster 4 *n* = 6). **d** Proportion of PAM50 ROR high, medium and low score in each DNADX CN-based subtype (Cluster 1 *n* = 17, Cluster 2 *n* = 15, Cluster 3 *n* = 11, Cluster 4 *n* = 6). **e** Proportion of high and low CelTIL responders after one dose of HER3-DXd in each DNADX CN-based subtype (Cluster 1 *n* = 17, Cluster 2 *n* = 15, Cluster 3 *n* = 11, Cluster 4 *n* = 6). *P*-values (p) in (**c**–**e**) were determined by two-sided Fisher's exact test. Source data are provided as a Source Data file.

Across these 49 patients with DNA-sequencing data, HER2DX *ERBB2* expression was found moderately correlated with the copy-number signal of chromosome 17q12, where *ERBB2* gene is located (Pearson correlation coefficient = 0.55, *p* < 0.001) (Supplementary Fig. 14a). Concordant with this finding, high copy-number signal of 17q12 was also found inversely associated with CelTIL response as a continuous variable (*p* = 0.002; AUC = 0.797) and as group categories defined by tertiles (*p* = 0.007; Supplementary Fig. 14b). For example, the proportion of patients with a CelTIL response in 17q12 CN segment low-tertile, medium-tertile and high-tertile groups was 56.3%, 35.3% and 6.3%, respectively (Supplementary Fig. 14c). Finally, we evaluated the performance of the 2 CN-based HER2-related signature scores as continuous variables to predict Cel-TIL response after one dose of HER3-DXd. The ROC AUC values for the CN-based HER2-IHC signature (which predicts HER2 protein expression) and the CN-based HER2 amplicon signature (which predicts *ERBB2* amplifications) were 0.812 and 0.843, respectively (Supplementary Fig. 14b). As categorical variables defined by tertiles, the CN-

based HER2-IHC signature and the CN-based HER2 amplicon signature were also significantly associated with CelTIL response. CelTIL response in CN-based HER2-IHC signature-low, -medium, and -high groups were 59%, 25%, and 12%, respectively. CelTIL response in CN-based HER2 amplicon signature-low, -medium and -high groups were 65%, 19%, and 12%, respectively (Supplementary Fig. 14c).

## CN-based molecular subtypes and CelTIL response
Using unsupervised analysis from 150 CN-based signatures, we have recently identified 4 DNADX subtypes (i.e., clusters -1, -2, -3, and -4) with prognostic value within HR+/HER2- breast cancer[18]. Across 49 tumors from SOLTI-1805 TOT-HER3 trial part A with DNA-sequencing data, the proportion of patients with DNADX cluster-1 disease was 34.7%, cluster-2 was 30.6%, cluster-3 was 22.5%, and cluster-4, 12.2% (Fig. 5a). As expected, DNADX cluster-1 had high scores of Luminal A-like signatures, DNADX cluster-2 had high scores of estrogen-related and Luminal B-like signatures, DNADX cluster-3 had high scores of the Basal-like biology and proliferation-related signatures, and

DNADX cluster-4 had high scores of HER2, Luminal B-like and proliferation signatures (Fig. 5b). Concordantly, all PAM50 Basal-like tumors (n = 3) were identified as being part of DNADX cluster-3 (Fig. 5c), and no PAM50 ROR-low tumor (n = 5) was identified as being part of DNADX cluster-3 (Fig. 5d). To provide some more insight into the biology of the 4 DNADX subtypes, we assessed the distribution of the DNADX subtypes across the integrative clusters (IntClust 1-10) of the METABRIC cohort[19]. There were significant differences in the distribution of DNA subtypes (clusters 1-4) (chi-square p < 0.001). Higher representation of DNA-based clusters 3 and 4 were observed in the integrative clusters of worse prognosis (i.e: IntClust 5, 9, and 10), while the DNA-based clusters 1 and 2 were more identified in those integrative clusters of better prognosis (i.e. IntClust 3, 4, 7, 8) (Supplementary Fig. 15).

Finally, we evaluated the association of the 4 DNADX CN-based subtypes with CelTIL response. Overall, the 4 DNADX subtypes were found associated with CelTIL response (p < 0.001). The proportion of patients with CelTIL response across the 4 DNADX subtypes was 90.9% (cluster-3), 33.3% (cluster-4), 17.6% cluster-1, and 6.7% cluster-2 (Fig. 5e). Cluster-3 was found significantly associated with CelTIL response compared to clusters 1-2-4 (84.2% vs. 9.1%; odds ratio = 53.33, p < 0.001) and this association was found independent of HER2DX *ERBB2* mRNA expression (odds ratio = 20.13, p = 0.040). In this bivariate analysis, HER2DX *ERBB2* mRNA was also independent of DNADX cluster 3 (odds ratio = 0.04, p = 0.008).

## Discussion

In the last few years, the introduction of ADCs has changed the treatment paradigm of breast cancer. However, identifying biomarkers for optimal patient selection remains an unmet need[20,21]. The efficacy of some ADCs correlates with the expression levels of their target (i.e: trastuzumab emtansine [T-DM1][22,23], an ADC targeting HER2; or sacituzumab govitecan[24], an ADC targeting TROP-2); and although some preclinical studies have suggested that response to HER3-DXd may correlate to HER3 expression in cancer cells[25,26], the clinical activity of HER3-DXd is observed across a wide range of HER3 expression levels[5–7]. Here, we evaluated RNA and DNA-based determinants of response to preoperative HER3-DXd in patients with newly diagnosed HR+/HER2- breast cancer using samples from the SOLTI-1805 TOT-HER3 trial[27].

The primary endpoint of the SOLTI-1805 TOT-HER3 trial was CelTIL score after one single dose[27], a biological surrogate of response based on the percentage of tumor cellularity and the percentage of stromal TILs[11,12,27]. In SOLTI-1805 TOT-HER3 part A, an association between clinical response and CelTIL change was observed[7] which has been validated in part B. More importantly, although CelTIL validation as a biomarker is not at level of evidence 1, our prior investigations suggest that the CelTIL score can serve as an early predictor of drug activity[11–13]. Indeed, this early biological response has been strongly linked to achieving a pathological complete response (pCR) after completing neoadjuvant therapy. For example, in HER2-positive breast cancer, an increase in CelTIL at day 15 following anti-HER2 neoadjuvant therapy correlated with pCR in both the PAMELA and LPT109096 phase II studies[11]. Additionally, the NeoALTTO phase III trial demonstrated that the same rise in CelTIL at day 15 was associated with improved 5-year event-free survival and overall survival[12]. Here, we also showed that an increase in CelTIL at day 21 following neoadjuvant eribulin monotherapy in HER2-negative breast cancer was significantly associated with RCB 0/1 after the completion of 4 cycles. Collectively, these findings underscore the value of the CelTIL score as an early indicator of ADC activity, and potentially an early indicator of long-term drug efficacy.

Among biomarkers of CelTIL response after one dose of HER3-DXd, we identified known determinants of response to standard chemotherapy such as high expression of basal-like and proliferation-related genes[28,29], low expression of luminal A and CES signatures[30,31] mutations in *TP53*[32]. Accordingly, CN-based signatures tracking basal-like, proliferation and luminal biology[17,18] were associated with response to HER3-DXd. Our results are consistent with preclinical evidence in breast cancer PDX suggesting that the basal-like subtype and *TP53* mutations are potential biomarkers of response to HER3-DXd[16]. Moreover, the newly identified DNADX DNA-based subtypes were also associated with CelTIL response. These subtypes have shown prognostic value in patients with early-stage and metastatic HR+/HER2- breast cancer and predict response to endocrine therapy in combination with CDK4/6 inhibition in patients with metastatic HR+/HER2- breast cancer[18]. Among the 4 DNADX subtypes, the so-called DNADX cluster-3, which is associated with poor response to endocrine therapy plus CDK4/6 inhibition[18] and increased response to HER3-DXd, shows low features of luminal-related biology and high features associated with the basal-like biology such as high proliferation and *RB1* loss-of-heterozygosity.

Intriguingly, high HER2 protein and HER2DX *ERBB2* mRNA expression, high expression of RNA/DNA-based HER2-related signatures and high CN signal of 17q12 (where *ERBB2* is located) were associated with low CelTIL response. Our initial hypothesis posited that lower *ERBB2* levels might be predictive of an enhanced CelTIL response in HER2-negative disease. However, subsequent analyses reveal a more intricate interplay, meriting deeper exploration. We hypothesize, based on the known heterodimerization of HER2-HER3[33–35], that in contexts of high HER2 expression, HER2-HER3 dimers may be more stable potentially diminishing HER3-DXd internalization and efficacy, yet further investigation is warranted. These insights contribute to the evolving narrative that *ERBB2* levels are not merely a biomarker but may actively influence the pharmacodynamics of HER3-DXd, shedding light on the variability in response observed. This hypothesis underscores the complex biological orchestra governing drug sensitivity and resistance, paving the way for more targeted and effective therapeutic strategies.

Interestingly, expression of *ERBB2* levels was also found to be higher in PDX models that relapsed to HER3-DXd compared to non-relapsed models[16]. Conversely, we have demonstrated that high HER2DX *ERBB2* is a biomarker of sensitivity to the HER2-targeting ADC T-DM1 in patients with advanced HER2-positive breast cancer[22,36]. In HR+/HER2- breast cancer, both T-DXd and HER3-DXd, ADCs directed against HER2 and HER3 respectively, have shown efficacy. Indeed, T-DXd has recently been approved by the FDA for the treatment of patients with HER2-low metastatic breast cancer[37] and it is currently under investigation in the neoadjuvant[38] and adjuvant[39] settings. Based on our results, we hypothesize that, within HER2-negative disease, tumors with very low *ERBB2* levels may benefit more from HER3-DXd than T-DXd, and vice-versa, in *ERBB2*-high tumors. Indeed, in patients with metastatic breast cancer treated with HER3-DXd[6], PFS in HR+/HER2-low (5.5 months) and TNBC HER2-low (4.4 months) was shorter than in HR+/HER2-zero (8.2 months) and TNBC HER2-zero (8.4 months)[40]. On the contrary, a biomarker analyses from patients from DAISY trial demonstrated correlation between HER2 expression and T-DXd uptake[41], and PFS also differed significantly in the HER2-positive, HER2-low, and HER2-zero cohort with median PFS of 11.1, 6.7, and 4.2 months, respectively[42].

Our study has some limitations. First, patients with early-stage HR+/HER2- breast cancer in the SOLTI-1805 TOT-HER3 trial received only one dose of HER3-DXd; therefore, it is unknown if the determinants of response identified in this study will be validated when patients receive more doses such as a full neoadjuvant course of treatment. Second, we did not evaluate biomarkers of clinical response as this was only available for 62 patients and its measure was not standardized across participating centers of the SOLTI-1805 TOT-HER3 trial. Third, CelTIL score, an early read-out of drug activity and the primary endpoint of the SOLTI-1805 TOT-HER3 trial, may not always be associated with

clinical response. Fourth, DNA was only available for 49 samples of the SOLTI-1805 TOT-HER3 part A, and was not available in part B, however CelTIL score was comparable in the RNA and DNA cohorts. Fifth, HER2 IHC was assessed locally at each investigative site which may have contributed to underperformance of HER2 IHC in comparison with genomics. Sixth, we did not investigate the mechanism by which HER2 expression may prevent response to HER3-DXd and further translational and mechanistic studies are warranted. Seventh, due to low number, we did not perform subtype-specific analysis.

Most of these limitations will be overcome with the translational analysis of the ongoing SOLTI-2103 VALENTINE trial (NCT05569811), a parallel, non-comparative, three-arm, randomized open-label, multi-center study in 120 women or men with primary operable HR+/HER2- breast cancer with Ki67 ≥ 20% and/or high genomic risk (defined by gene signature) which was designed following the results of the SOLTI-1805 TOT-HER3 trial. The new study aims to evaluate the clinical benefit and biological effects of HER3-DXd with/without letrozole vs chemotherapy as a neoadjuvant treatment regimen. The primary aim is to evaluate the ability of each treatment strategy to achieve a pCR at surgery. Patients will undergo a baseline and day 21 biopsy. The translational correlative analysis will include the evaluation of CelTIL response and clinical response at surgery and the validation of the predictive biomarkers investigated in the current study.

In conclusion, HER3-DXd is a promising anti-cancer compound for early-stage HR+/HER2- breast cancer. Potential biomarkers for patient selection include RNA- and DNA-based signatures of basal-like, proliferation and luminal biology, cluster-3 of the DNA-based 4-subtype classification, and the expression of *ERBB2*.

## Methods

### SOLTI-1805 TOT-HER3 part A and B trial design

The primary results of the SOLTI-1805 TOT-HER3 part A study have been reported elsewhere[7]. Part B study is a prospective, multicenter, single-arm window of opportunity trial designed to assess the effects of a single pre-operative dose of 5.6 mg/kg of HER3-DXd in patients with early-stage HR+/HER2- or TNBC who have not undergone prior treatment. Patient selection criteria include histologically confirmed, non-metastatic, operable breast cancer with a primary tumor size of at least 1 cm, along with specific performance status and Ki67 expression requirements. Baseline assessments, including physical examination, radiological imaging, and biomarker analysis, were conducted to confirm eligibility. ER, PR, and HER2 statuses were determined locally. All enrolled patients received a single intravenous dose of HER3-DXd on day 1 of the treatment cycle. Mandatory biopsies were performed at day 21. The primary objective of part B was to evaluate the change in CelTIL score between baseline and day 21 tumor samples. Secondary objectives included change in CelTIL score according to baseline expression levels of *ERBB3* mRNA, safety and tolerability. Safety assessments were conducted using standardized criteria, and clinical response was evaluated by ultrasound. Following the study treatment, patients received subsequent treatment based on the investigator's discretion, which could include additional neoadjuvant systemic therapy and/or surgical resection. Post-operative locoregional and systemic treatments were administered according to local guidelines. The protocol for this window-of-opportunity trial (NCT04610528) including predefined primary and secondary endpoint is available at https://clinicaltrials.gov/study/NCT04610528 and in Supplementary Note 1. Written informed consent was obtained from all study participants before the initiation of any study-specific assessments. This trial was conducted in compliance with the protocol, regulatory requirements, an independent ethics committee in accordance with the International Council for Harmonisation of Technical Requirements for Pharmaceuticals for Human Use guidelines for Good Clinical Practice, and the ethical principles of the latest revision of the Declaration of Helsinki as adopted by the World Medical Association

and approved by the independent ethics committee of Hospital Clínico de Valencia and the the Spanish Agency for Medicines and Health Products.

### Characteristics of patients included in the translational study

SOLTI-1805 TOT-HER3 is a single-arm multi-center window-of-opportunity trial (NCT04610528) for patients with newly diagnosed early-stage HR+/HER2- breast cancer performed across 10 sites in Spain[27]. Patients were pre- or post-menopausal, and tumors had a minimum tumor size of 1.0 cm and a Ki67≥10% by local assessment. In part A of the trial, 78 patients were allocated prospectively in 4 groups based on the tumor's *ERBB3* mRNA levels (i.e., ultra-low, low, high and ultra-high)[7]. As the CelTIL score was not evaluated in the day 21 (C1D21) biopsy from one patient, the analysis was performed on a total of 77 individuals. Patients received a single dose of HER3-DXd (6.4 mg/kg), and a mandatory tumor biopsy was performed at C1D21. The primary objective of the trial was to evaluate changes in the CelTIL score[11,12]. After that, patients underwent standard therapy (i.e., primary surgery or neoadjuvant therapy) at the physician's discretion. In Part B, a lower dose of HER3-DXd (5.6 mg/kg) was evaluated in 20 and 17 patients with HR+/HER2- and TNBC, respectively, without a pre-selection based on *ERBB3* mRNA baseline levels. TNBC samples from part B were not evaluated in the translational study. Baseline data and CelTIL response data are included in Supplementary Data 1 (part A) and Supplementary Data 5 (part B).

### Tumor sample characteristics

FFPE tumor samples were collected at baseline and C1D21 (Fig. 1). Tumor cellularity and the proportion of TILs were scored in whole sections of tumor tissue stained with hematoxylin and eosin (H&E). Tumor cellularity was defined as the percentage of tumor cells to the tumor area. TILs were quantified according to International TILs Working Group Guidelines[43,44]. CelTIL was scored as -0.8 × tumor cellularity (in %) + −1.3 × TILs (in %) and scaled to reflect a range from 0 to 100 points[11]. High CelTIL scores identify tumors that are highly immune infiltrated with reduced tumor cellularity. CelTIL score reported in TOT-HER3 was determined by 1 breast pathologist. However, for validation purposes, a second pathologist also evaluated CelTIL score in baseline samples of TOT-HER3. The correlations coefficients among the 2 pathologists were: % TILs Cor=0.846, p<0.001, % cellularity Cor=0.784, p<0.001, CelTIL Cor=0.800, p<0.001 (Supplementary Fig. 16). HER2 IHC and in situ hybridization were performed locally following according to the American Society of Clinical Oncologists/College of American Pathologists guidelines[45].

### Gene expression analysis

RNA was extracted from baseline pre-treatment FFPE tumor samples using the High Pure FFPET RNA isolation kit (Roche, Indianapolis, IN, USA). One to five 10-μm FFPE slides depending on tumor cellularity were used for each tumor sample, and macrodissection was performed, when needed, to avoid normal tissue contamination. A minimum of 100 ng of total RNA was analyzed on the nCounter platform[46] (Nanostring Technologies, Seattle, USA) using a 192-gene custom panel[47], which includes the 50 genes of the PAM50 test, the 27 genes of the HER2DX assay[47], immune-related genes, breast cancer-related genes including *ERBB3* and 7 housekeeping genes (*ACTB, MRPL19, GAPD, PSMC4, PUM1, RPLP0,* and *SF3A1*). Counts for each tumor are provided in Supplementary Data 2 (part A) and Supplementary Data 6 (part B). For each sample, research-based PAM50 subtyping was performed[48], and the scores of the PAM50 signatures including the Luminal A, Luminal B, HER2-enriched, Basal-like, Normal-like, the Proliferation score, ROR score, and the CES score[31] were also calculated. Finally, we evaluated the scores of the 4 HER2DX signatures (i.e., B-cell immune immunoglobulin (IGG), luminal, proliferation, and HER2 amplicon) and HER2DX *ERBB2* score[47].

## DNA-sequencing analysis

DNA was obtained from 49 baseline pre-treatment FFPE tumor samples using the QIAamp DNA FFPE Tissue kit (QIAGEN Inc.). A minimum of 100 ng of DNA was processed for library preparation using a custom hybridization-based capture panel targeting 435 genes with reported somatic mutations in different tumor types (VHIO-300 v4 panel) performed with Agilent SureSelectXT Low Input Target Enrichment System (Agilent Technologies, Inc). Indexed libraries were quantified by qPCR using the KAPA Library Quantification Kit (Roche Sequencing Solutions), pooled and sequenced in a HiSeq 2500 Illumina (2 × 100bp) at an average coverage of 500x. Reads were aligned to the hg19 reference genome with BWA[49], applied GATK[50] base quality score recalibration, indel realignment and duplicate removal. Variant calling (VarScan2 v2.4.3) required a minimum of 7 reads supporting the variant allele to call a mutation based on an internal analytical validation of the VHIO-300 panel in 121 FFPE samples. The sensitivity of the technique is 5% mutant allele frequency (MAF) for single nucleotide variants and 10% MAF for INDELs. Frequent single nucleotide polymorphisms (SNPs) in the population were removed based on the gnomAD database (allele frequency ≤ 0.0001). Mutations for each tumor are provided in Supplementary Data 4. Copy numbers (CN) were calculated from an in-built genome-wide SNP backbone targeting 20000 SNPs using CNVkit (v0.9.6.dev). Data were manually curated, and classification of identified variants was performed using publicly available databases (COSMIC, cBioPortal, ClinVar, VarSome, OncoKB).

## DNA-based signatures

A total of 150 DNA-based phenotypic signatures, and 4 DNA-based subtypes (clusters-1, -2, -3, and -4), were identified[17,18]. Briefly, DNA-sequencing segmentation files from CNVkit output (for tumor DNA) were first mapped to gene-level features. The signal of 519 DNA segments was calculated using the mean copy number score across genes within each segment. The coefficients of DNA segments for predicting gene signatures were obtained from Xia et al. DNA-based signature scores were calculated as the weighted average of DNA segment values for each sample: a final signature score was obtained by adding all values (i.e., coefficient of segment A × signal of segment A plus coefficient of segment B × signal of segment B…)[17]. The 4 DNA-based subtypes or clusters were identified using a previously reported DNA-based subtype predictor, which is based on unsupervised analysis of tumor samples and the 150 DNA-based signatures[18]. For the 49 samples with the 150 DNA-based signatures available, we calculated the Euclidean distances to the 4 centroids and assigned a cluster class to each sample based on the nearest centroid. Signatures and clusters for each tumor are provided in Supplementary Data 3.

## Other clinical and PDX cohorts

The SOLTI-1007-NeoEribulin was a phase II, open-label, two-cohort, exploratory study in 174 patients with clinical stage I–II HER2-negative breast cancer receiving neoadjuvant eribulin monotherapy treatment[14]. CelTIL at baseline and C2D1, pCR, RCB and *ERBB2* mRNA data were evaluated. Additionally, gene expression and pCR data from 490 patients with HER2-negative breast cancer treated with neoadjuvant taxane/anthracycline-based chemotherapy[15] was downloaded from the Gene Expression Omnibus (GEO) under the accession number GSE25066. The *ERBB2* mRNA expression of a cohort of 147 patients with early-stage HER2-positive breast cancer treated at Hospital Clínic Barcelona[47] was interrogated. This examination aimed to provide insights into the treatment outcomes for this specific patient group. Data from the METABRIC study[19], including IntClust and copy-number data, was sourced from the cBioportal[51]. Processed DNA segment values were downloaded, DNA-based signature scores were calculated as the weighted average of DNA segment values for each sample, and DNA subtypes were determined as for Prat et al.[18]. IntClust and DNA-based signature were available for a 1679 tumor samples. Finally, *ERBB2* mRNA was interrogated in a cohort of 30 breast cancer PDX models treated with HER3-DXd[16].

## Statistics and reproducibility

To evaluate the association between each baseline variable and CelTIL response at C1D21, we utilized both uni- and multi-variable logistic regression models and conducted an analysis of the areas under the receiver-operator curve (ROC AUC). In addition, we performed unpaired and multiclass significance analysis of microarrays (SAM) using a false discovery rate (FDR) <10% to identify differential gene expression between CelTIL response vs. no-response. CelTIL response vs. no-response was defined as an absolute increase of CelTIL of ≥20 points between the two time-points (i.e., C1D21 minus baseline). Other cutoffs to define CelTIL response (i.e., absolute increases of CelTIL of >0, ≥10, ≥20, ≥30 and ≥40) were also evaluated with very similar results. An absolute increase of CelTIL of ≥20 points predicted clinical response with an AUC of 0.68 ($p = 0.009$) and identified a higher number of significant genes compared to lower cutoffs. Two-sided $p$-values < 0.05 were considered statistically significant. The 20 HR +/HER2− tumor samples from part B were only used for independent validation of RNA-based findings in part A. Statistical computations were carried out in R 4.0.3 (http://cran.r-project.org). No data were excluded from the analyses.

## Reporting summary

Further information on research design is available in the Nature Portfolio Reporting Summary linked to this article.

## Data availability

The protocol of the SOLTI-1805 TOT-HER3 study is available as a Supplementary file (Supplementary Note 1). The data generated in this study including gene expression counts, PAM50 subtypes, DNA signatures and subtypes, and pathological data can be found in Supplementary Material. FASTQ and BAM files from targeted DNA-Seq experiments of 49 breast cancer samples have been submitted to EGA European Genome-Phenome Archive under the accession number EGAD50000000562. Data are available under restricted access as participants of this study did not agree for their sequencing data to be shared publicly. Access can be obtained for academic use only, under a data transfer agreement and upon Ethics Committee approval. The timescale for this process is approximately 6 months and the data will be available for 3 years. GSE25066 gene expression and pCR data were downloaded from the Gene Expression Omnibus (GEO) under the accession number GSE25066. METABRIC data were downloaded from cBioportal [https://www.cbioportal.org/study/summary?id=brca_metabric]. The data generated in this study and presented in the figures are provided in the Supplementary Data/Source Data files. Source data are provided with this paper.

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

## Acknowledgements

This study was sponsored by SOLTI Cancer Research Group and supported by Daiichi Sankyo, Inc. F.B.M. received funding from Fundación científica AECC Ayudas Investigador AECC 2021 (INVES21943BRAS). O.M.S. is a 2022 SEOM fellow. AP received funding from Fundación CRIS contra el cáncer PR_EX_2021-14, Agència de Gestó d'Ajuts Universitaris i de Recerca 2021 SGR 01156, Fundación Fero BECA ONCOXXI21, Instituto de Salud Carlos III PI22/01017, Asociación Cáncer de Mama Metastásico IV Premios M. Chiara Giorgetti, Breast Cancer Research Foundation BCRF-23-198, and RESCUER, funded by European Union's Horizon 2020 Research and Innovation Programme under Grant Agreement No. 847912. V.S. received funding from Asociación Cáncer de Mama Metastásico III Premios M. Chiara Giorgetti. Supplementary Fig. 11 was produced by Antonio García, a scientific illustrator from Bio-Graphics.

## Author contributions

F.B.M., J.M.F.C., A.P., and M.O. designed the study. F.B.M., J.M.F.C, C.F., O.M.S., J.M.C., M.M., P.T., F.J.S.B, J.C., B.G.F., E.S., A.O, V.S., F.P., A.M.L.B, M.A., J.A.G, G.V., R.S.B, E.C., M.E.B., Y.I., P.G., J.M., S.P., M.V., A.S., D.S., S.E., P.D.F., F.S., A.V., T.P., A.P., and M.O. contributed to data collection and assembly. F.B.M., J.M.F.C., C.F., B.G.F., E.S., A.O, V.S., A.V., T.P., A.P., and M.O. interpreted and analyzed the data. All authors wrote and reviewed the report and approved the final version for submission.

## Competing interests

Potential conflicts of interest are the following: F.B-M. reports patents filed: PCT/EP2022/086493, PCT/EP2023/060810, EP23382703 and EP23383369, and part time employment by Reveal Genomics. C.F. is currently employed by AstraZeneca. O.M-S. reports travel expenses and consulting fees from Roche, AstraZeneca and Reveal Genomics and speaker fees from Eisai, Novartis and Daiichi. J.M.C. reports travel expenses and speaker fees from AstraZeneca, Lilly, Pfizer, Gilead, Novartis and MSD. R.S-B. has received travel grants from Pfizer, Astra Zeneca, and Novartis, and honoraria for speaker or advisory board participation from Novartis, Lilly, Astra Zeneca, Daiichi Sankyo, Roche, Glaxo Smith Kline, Clovis Oncology, Seagen, and Accord. R.S.-B. non-financial interests include European Society of Medical Oncology Young Oncologists Committee member, Spanish Society of Medical Oncology —Scientific Secretary. G.V. has received a speaker's fee from MSD,

Pfizer, GSK and Pierre Fabrer, has held an advisory role with AstraZeneca and received consultant fees from Reveal Genomics. E.C. reports advisory and consulting fees or speaker honoraria from Roche, Pfizer, Lilly, AstraZeneca, Daiichi Sankyo, Menarini, MSD, Novartis, Gilead, and Reveal Genomics, institutional financial interests from Roche, Pfizer, travel grants from Roche, Pfizer, AstraZeneca, Daiichi Sankyo, and non-financial steering committee member for Roche, Astra Zeneca, Daiichi Sankyo, Novartis, Reveal Genomics. S.P. reports travel grants from Gilead, Roche, Astra-Zeneca, and Pfizer, and honoraria for speaker or advisory board participation from SeaGen, Novartis, Lilly, Pfizer, Astra Zeneca-Daiichi, Roche, and Gilead. J.C, reports speaker honoraria or advisory and consulting fees from Glaxo, AstraZeneca, Roche, Novartis, Pharmamar, Eisai, Lilly, Pierre Fabre, Daichii Sankyo, Seagen, Deciphera and Pfizer and travel grants from Gilead, Astra Zeneca, Daichii Sankyo, Roche, Novartis, Pharmamar. A.P. reports consulting fees from Roche, Novartis, AstraZeneca, Daiichi-Sankyo, and Peptomyc; patents filed PCT/EP2016/080056, PCT/EP2022/086493, PCT/EP2023/060810, EP23382703 and EP23383369; stockholder and consultant of Reveal Genomics; and institutional financial interests from Roche, Novartis, AstraZeneca, Daiichi-Sankyo, Reveal Genomics, Ona Therapeutics, BMS, and Pfizer. M.O. reports institutional grant/research from AstraZeneca, Ayala Pharmaceuticals, Boehringer-Ingelheim, Genentech, Gilead, GSK, Immutep, Roche, Seagen, Zenith Epigenetics, advisory and consulting fees or speaker honoraria from AstraZeneca, Cureos Science, Daiichi-Sankyo/AstraZeneca, Gilead, iTEOS, Lilly, MSD, Relay Therapeutics, Roche, Seagen, Eisai, Libbs, Novartis, Pfizer, travel grants from Astra-Zeneca, Eisai, Gilead, Pierre-Fabre, and non-financial disclosure as the SOLTI Breast Cancer Group president elected. The remaining authors declare no competing interests.

## Additional information

Fara Brasó-Maristany ●[1,2,3,4], Juan Manuel Ferrero-Cafiero ●[2,5], Claudette Falato[1,2], Olga Martínez-Sáez ●[1,2,4], Juan Miguel Cejalvo[2,6,7], Mireia Margelí[2,8], Pablo Tolosa[2,9], Francisco Javier Salvador-Bofill[2,10], Josefina Cruz[2,11], Blanca González-Farré[1,2,12], Esther Sanfeliu[1,2,12], Andreu Òdena ●[5,13], Violeta Serra ●[13], Francisco Pardo ●[1,3], Ana María Luna Barrera[14], Miriam Arumi[2,15,16], Juan Antonio Guerra[17], Guillermo Villacampa[2], Rodrigo Sánchez-Bayona[2,9], Eva Ciruelos[2,9], Martín Espinosa-Bravo ●[2,18], Yann Izarzugaza[2,19], Patricia Galván[1,2,3], Judith Matito[20], Sonia Pernas ●[2,21,22], Maria Vidal ●[1,2,4], Anu Santhanagopal[23], Dalila Sellami[23], Stephen Esker[23], Pang-Dian Fan[23], Fumitaka Suto ●[23], Ana Vivancos ●[20], Tomás Pascual ●[1,2,4], Aleix Prat ●[1,2,3,4,24,25] ✉ & Mafalda Oliveira[2,15,16]

[1]Translational Genomics and Targeted Therapies in Solid Tumors, August Pi i Sunyer Biomedical Research Institute (IDIBAPS), Barcelona, Spain. [2]SOLTI Cancer Research Group, Barcelona, Spain. [3]Reveal Genomics, Barcelona, Spain. [4]Cancer Institute and Blood Diseases, Hospital Clinic de Barcelona, Barcelona, Spain. [5]University of Barcelona, Barcelona, Spain. [6]Medical Oncology Department, Hospital Clínico Universitario de Valencia, Valencia, Spain. [7]Breast Cancer Biology Research Group, Biomedical Research Institute INCLIVA, Valencia, Spain. [8]Medical Oncology Department, ICO - Institut Català d' Oncologia Badalona (Hospital Universitario Germans Trias i Pujol), Badalona, Spain. [9]Medical Oncology Department, Hospital 12 de Octubre, Madrid, Spain. [10]Medical Oncology Department, Hospital Universitario Virgen del Rocio, Sevilla, Spain. [11]Medical Oncology Department, Hospital Universitario de Canarias, Santa Cruz de Tenerife, Spain. [12]Pathology Department, Hospital Clinic of Barcelona, Barcelona, Spain. [13]Experimental Therapeutics Group, Vall d'Hebron Institute of Oncology (VHIO), Barcelona, Spain. [14]Centro Integral Oncológico Clara Campal HM (CIOCC), Madrid, Spain. [15]Medical Oncology Department, Vall d'Hebron University Hospital, Barcelona, Spain. [16]Breast Cancer Group, Vall d'Hebron Institute of Oncology (VHIO), Barcelona, Spain. [17]Medical Oncology Department, Hospital de Fuenlabrada, Madrid, Spain. [18]Breast Cancer Surgical Unit, Vall d' Hebron University Hospital, Barcelona, Spain. [19]Medical Oncology Department, Fundación Jimenez Díaz, Madrid, Spain. [20]Cancer Genomics Group, Vall d'Hebron Institute of Oncology (VHIO), Barcelona, Spain. [21]Bellvitge Biomedical Research Institute IDIBELL, L'Hospitalet de Llobregat, Barcelona, Spain. [22]Medical Oncology Department, Institut Català d'Oncologia, L'Hospitalet de Llobregat, Barcelona, Spain. [23]Research and Development, Daiichi Sankyo, Inc, Basking Ridge, NJ, USA. [24]Department of Medicine, University of Barcelona, Barcelona, Spain. [25]Institute of Oncology (IOB)-Hospital Quirónsalud, Barcelona, Spain. ✉e-mail: alprat@clinic.cat

