## [Peer Review File · Nature Communications]

REVIEWER COMMENTS

Reviewer #1 (Remarks to the Author): Expert in breast cancer targeted therapy, clinical and translational research, and molecular genetics

Brasó-Maristany and colleagues examine potential genetic determinants of response to HER3-DXd in patients with untreated early-stage HR+/HER2- breast cancer. The primary endpoint of the trial is evaluation of changes in the CeTIL score, an assessment of tumor cellularity and tumor infiltrating lymphocytes. Performing RNA-based expression, the authors find high expression of proliferation and cell division-related genes such as AURKA, CCNE ($p=0.014$), and MKI67 was associated with CeTIL response. High expression of luminal-related genes such as NAT1, SLC39A6, MAGED2, and THSD4 was associated with lack of CeTIL response. Interestingly, low HER2 levels are associated with CeTIL response. TP53 mutations are associated with high CeTIL response after one dose of HER3-DXd. Examination of DNA copy-number based signatures associated with CeTIL response found scores of signatures related to the Basal-like biology and high proliferation were associated with CeTIL response. The same group has recently identified 4 subtypes (clusters -1, -2, -3, and -4) with prognostic value within HR+/HER2- breast cancer examining circulating tumor DNA. Cluster-3 was found significantly associated with CeTIL response compared to clusters 1-2-4 and is associated with basal-like biology and proliferation-related signatures. Comments, concerns, and limitations are detailed below.

- The primary endpoint of the trial is evaluation of changes in the CeTIL score and as such the manuscript examines DNA and RNA correlates in the context of CeTIL score. The authors argue that their prior investigations suggest that the CeTIL score can serve as an early predictor of drug activity. However, CeTIL score may not always predict drug activity, and this is a limitation of the work.
- Somewhat surprising is the finding that low HER2 levels is associated with a CeTIL response. The authors postulate that within HER2-negative disease, tumors with very low ERBB2 levels may benefit more from HER3-DXd. They acknowledge a limitation is lack of investigating the mechanism by which HER2 expression may prevent response to HER3-DXd and further translational and mechanistic studies are warranted. At a minimum, further discussion and a hypothesis on why increased HER2 expression would result in diminished response to HER3-DXd.
- An additional limitation of the work is the discordance with data presentation that have been previously published. It would be helpful to correlate HER3 expression level with HER2 levels.

Minor: Grammar edit of abstract

Reviewer #2 (Remarks to the Author): Expert in breast cancer targeted therapy, molecular genetics, and signalling

This work examined potential RNA and DNA-based determinants of response to preoperative HER3-DXd, an HER3-targeted ADC, in patients with untreated early-stage HR+/HER2- breast cancer. The CeTIL score, defined as the variation in tumor cellularity and tumor-infiltrating lymphocytes between baseline and

day 21 upon HER3-DXd treatment, was used to evaluate the drug response. It seemed that determinants of chemosensitivity and low ERBB2 expression might be potential biomarkers of early response to HER3-DXd in patients with HR+/HER2- breast cancer.

Specific concerns:

1. Previous investigations suggest that the CelTIL score may serve as an early predictor of drug activity. The current study used CelTIL score only to evaluate the response to HER3-DXd. However, additional methodology should be applied to confirm the findings.
2. HER3-DXd has been tested in clinical trials of other cancer patients, including the NSCLC patients who have developed resistance to EGFR-TKI. It is not clear how to compare the current studies with the other clinical trials in terms of the drug dose and efficacy.
3. HER3-DXd was tested in the patients with early-stage HR+/HER2- breast cancer. It would be interesting to know how many of those are ER+ and how many are PR+. It is not clear if ER and PR expression may make any difference in the response to HER3-DXd.
4. Low ERBB2 expression has been found to be a potential biomarker of early response to HER3-DXd in the patients with HR+/HER2- breast cancer. This observation has been discussed in recognition of the study's limitations. However, the underlying mechanism is worth discussing further.

Reviewer #3 (Remarks to the Author): Expert in breast cancer genomics, subtypes, and biomarkers; co-reviewed with Reviewer #4

Summary of the work:

The manuscript by Brasó-Maristany et al. has investigated molecular features associated to HER3-dxd treatment response in HR+/HER2- breast cancer.

In this work, the authors have performed molecular characterization of HR+/HER2- baseline biopsies from the SOLTI-1805 TOT-HER3 trial and investigated how these molecular features associate to change in CelTIL scores to reveal predictors of drug response. CelTIL score is previously published.

Their concluding remark is: "Determinants of chemosensitivity, a new DNA-based subtype classification and low ERBB2 expression are potential biomarkers of early response to HER3-DxD treatment in HR+/HER2- BC2."

We have the following comments:

1. Introduction: This section mainly elaborates on the groups own works and the present trial. Please give some more general introduction into the rationale for this work.
2. Why is the ongoing trials SOLTI-2103 and VALENTINE relevant for introduction (line106)?
3. Material and methods: The description of the trial and the patient characteristics is tidy and easy to follow. It is good to see that the scoring of CelTIL has been done by a pathologist and that a correlation

test with a different pathologist is performed. As the CelTIL score is established for HER2+ breast cancer, is there a concern that the score has different range for HER2- samples and should be weighed differently in the score? Have the distribution of scores been compared between HER2+ and HER2- tumors? CelTIL scores is considered as surrogate markers of drug activity and early response. We note that while the association of response to CelTIL score is shown in neoadjuvant trials for HER2+ patients, the predictive power seems not to be not fully established in HER2- setting? Please elaborate, preferentially with data.

4. The authors justify the use of CelTIL scores as a surrogate to clinical response by stating that measurement of clinical response was not standardized and not available for all the patients. Are the authors disapproving the validity of clinical response assessment in the TOT-HER3 study? The published TOT-HER3 study states that 62 patients were assessable for clinical response; we find this to be a reasonable number of patients? What was the results if clinical response was used as a parameter instead of CelTIL score? (see line 267).

5. Results: The section “study design” contains information given partly both in Introduction and in Material and Methods, please avoid so many repetitions (start at line 133?).

6. The relationship between ERBB2 expression/amplification and CelTIL score: These tumors are by definition not having high ERBB2 protein expression, and the range of gene expression is not similar to a pan-subtype BC cohort, it might not be correct to refer to the tertiles as low, intermediate and high? How similar/dissimilar is the ERBB2 expression of these ‘high’ tumors with what is seen in HER2+ tumors? (see line 181 and onwards + Fig 3).

7. The translation of molecular signatures from discovery to clinical practice is extremely low due to challenges associated for instance with technical variabilities data processing methods. In the light of this fact, as the study finds 90 of 150 CN-based signature to be associated with CelTIL scores, the authors should clarify on overlap and association between these signatures.

8. Discussion: The discussion highlights the interesting finding concerning ERBB2 expression and CelTIL score and is also acknowledging the limitation of the study. The correlation between PAM50 subtypes and known, related parameters (such as TP53 mutation, proliferation score and ROR scores) and CelTIL score is important to highlight as this shows the need to have larger trials to be able to detect subtype-specific predictive biomarkers/signatures. We acknowledge that it is not always possible to perform subtype-specific analysis but this point should be considered while interpreting the results and could be a 4th point in discussion of limitations (from line 308).

9. Access to data: the molecular data are derived from Nanostring RNA analysis and DNA target panel sequencing and should not pose a risk for patient re-identification. We think the data would be very important to release, both for assessing the validity of the analyses but also for further explorations by other researchers. The data (including metadata) should be deposited at a commonly used repository where it can be accessible (also able to do it only upon request).

10. Conclusion: The final sentence in the abstract can be considered an overstatement: There are not shown that ERBB2low is predicting treatment response (only low CelTIL score), the DNA based classification is not new (but previously published, can be misunderstood). Please rephrase.

Minor comments:

1. Is the trial name TOTHER3 or TOT-HER3 (see line 106, 263 and onwards)

2. Figure 2: Will suggest to add in the legend that the expression is from baseline samples.

Reviewer #4 (Remarks to the Author): Expert in breast cancer genomics, subtypes, and biomarkers; co-reviewed with Reviewer #3

Reviewer #5 (Remarks to the Author): Expert in immuno-oncology and immunogenomics

Brasó-Maristany et. al. describe the results of 1 dose of neo-adjuvant Her3-ADC in breast cancer population to a translational composite endpoint CelTil. ADC's are an important re-emerging therapeutic and elucidating the biology behind their response is an important open question. This composite endpoint consists of both changes in tumor cellularity and changes in TILs and was previously shown to correlate to path complete response rate. It's important to note that in some sense, they are analyzing a surrogate (CelTil) of a surrogate pathologic complete response rate to an actual hard clinical endpoint (like OS or PFS). That in some sense, may be an important limitation in the biologic findings they are ultimately revealing and should be discussed more clearly in the discussion.

The manuscript overall is well written and easy to understand, but it remains unclear to me what the authors believe their primary finding is. They present a number of analyses, that seem to suggest that perhaps TNBC has a higher response rate (PAM50 subtype, increased TP53 mutations, and copy number signature data) – but don't come outright and say that; nor do they present receptor status data with response.

The other challenge with this manuscript is that the primary endpoint is a composite endpoint to some extent (TIL changes and tumor cellularity changes). It would be nice to understand if the findings with CelTil they ultimately find are driven by one of the two components or are in themselves composite findings? (this could be a final figure, exploring what the authors believe are their main findings).

1.) Lines 135-136 – what “clinical response” is being correlated with a CelTil change of ≥ 20 points with an AUC of 0.68? Is that data from this paper or a prior publication? No reference is cited.

2.) Methods to determine CelTil are missing from the methods section, presumably this is from H&E assessment as described in Nuciforo et. al, 2019?

3.) The authors show marked influence of Basal and Her2 subtypes in CelTil response in Supplementary Figure 1 – are these results robust to multiple testing correction? (OR of 19 and 10). These seem like main findings? However, contradictory to this result – the authors show in Figure 3B that low Her2 by

MRNA or IHC appears to be associated with higher response – in contrast to the PAM50 subtype finding. How do the authors reconcile these conflicting results?

4.) The description of the DNA-based copy number signatures in the method is not clear. Is this just whether certain segments of the genome are deleted or amplified at 150 different sites with clustering? Although the authors provide 2 references for this method – this is not a commonly used analysis, and should be described more clearly in the methods here.

5.) The HER2 amplicon data from copy number signature appears concordant with mRNA and IHC, but discordant with the PAM50 subtyping – what do the authors make of this? (Figure 5B)

6.) The CN clustering data in figure 6 would be more useful if it was clearly identified with previously described copy number clustering in breast cancer – either from TCGA or another large consortium project. Can the authors also include data on fraction of genome altered from the copy number analysis?

7.) The data availability statement is factually wrong. It is feasible to provide anonymized expression data along with clinical outcomes. This has been done by innumerable other papers on clinical trials previously. Ideally, at least some of the data should be publically available in repositories or supplementary tables.

Minor

- Figure 4A oncoprints just illustrate typical mutations for each subtype, this could be a supplementary figure

- Figure 4B p-value for TP53 difference in CelTil response is different in figure vs. text description (line 197-198 $p = 0.01$ vs $p=0.043$)

Response to reviewers

Reviewer #1 (Remarks to the Author): Expert in breast cancer targeted therapy, clinical and translational research, and molecular genetics

Brasó-Maristany and colleagues examine potential genetic determinants of response to HER3-DXd in patients with untreated early-stage HR+/HER2- breast cancer. The primary endpoint of the trial is evaluation of changes in the CelTIL score, an assessment of tumor cellularity and tumor infiltrating lymphocytes. Performing RNA-based expression, the authors find high expression of proliferation and cell division-related genes such as AURKA, CCNE ($p=0.014$), and MKI67 was associated with CelTIL response. High expression of luminal-related genes such as NAT1, SLC39A6, MAGED2, and THSD4 was associated with lack of CelTIL response. Interestingly, low HER2 levels are associated with CelTIL response. TP53 mutations are associated with high CelTIL response after one dose of HER3-DXd. Examination of DNA copy-number based signatures associated with CelTIL response found scores of signatures related to the Basal-like biology and high proliferation were associated with CelTIL response. The same group has recently identified 4 subtypes (clusters -1, -2, -3, and -4) with prognostic value within HR+/HER2- breast cancer examining circulating tumor DNA. Cluster-3 was found significantly associated with CelTIL response compared to clusters 1-2-4 and is associated with basal-like biology and proliferation-related signatures. Comments, concerns, and limitations are detailed below.

1. The primary endpoint of the trial is evaluation of changes in the CelTIL score and as such the manuscript examines DNA and RNA correlates in the context of CelTIL score. The authors argue that their prior investigations suggest that the CelTIL score can serve as an early predictor of drug activity. However, CelTIL score may not always predict drug activity, and this is a limitation of the work.

Thank you for your insightful comment. We appreciate the opportunity to provide further evidence supporting the value of the CelTIL score as a predictor of drug response in HER2-negative breast cancer, in addition to the existing data for HER2-positive (HER2+) cases^{1,2}.

Firstly, we wish to emphasize that the CelTIL score, which integrates tumor cellularity and tumor-infiltrating lymphocytes (TILs), serves as a biomarker of drug activity (which should be separated from drug efficacy). An increase in CelTIL indicates a reduction in tumor cellularity and an augmentation in TILs, and this effect, when it happens, is related to the drug's activity. In addition, our previous work in early-stage HER2+ breast cancer has demonstrated that there is a link between drug activity and drug efficacy (i.e., pCR status after neoadjuvant therapy and long-term survival outcome)^{1,2}.

In the SOLTI-1805 TOT-HER3 window-of-opportunity study (part A), a notable rise in CelTIL was observed in the majority of the 77 patients after a single HER3-DXd dose, correlating with tumor response at day 21³. Similarly, part B findings linked increased CelTIL scores to tumor response. These results affirm the score's potential as an early indicator of drug activity and possibly efficacy.

However, we acknowledge that the biomarker's validation is not at level of evidence 1, a limitation we have addressed in the manuscript's discussion section.

Furthermore, we now provide new evidence from the SOLTI-1007 NeoEribulin trial⁵, where patients with newly diagnosed HER2-negative breast cancer underwent 4 cycles of neoadjuvant eribulin monotherapy, and a FFPE biopsy was available before and at cycle 2 day 1 (C2D1). Here, CelTIL score increased after 1 eribulin cycle (at C2D1) in both HR+/HER2-negative and HR-/HER2-negative (triple-negative) breast cancers, with a subsequent rise at C2D1 statistically significantly associated with RCB 0/1 (vs. RCB 2-3) post 4 neoadjuvant eribulin cycles (p=0.013). This result strengthens the case for CelTIL's role as an early read-out of activity and efficacy following single agent chemotherapy (i.e., single agent chemotherapy, resembling HER3-DXd) in HER2-negative breast cancer. Of note, eribulin has weaker anti-proliferative and/or pro-apoptotic properties compared to HER3-DXd. We have now added these results in Supplemental Figure 1.

We believe these additional insights and data further substantiate the CelTIL score's potential and address the limitations noted. Thank you again for the opportunity to discuss these points.

- 2. Somewhat surprising is the finding that low HER2 levels is associated with a CelTIL response. The authors postulate that within HER2-negative disease, tumors with very low ERBB2 levels may benefit more from HER3-DXd. They acknowledge a limitation is lack of investigating the mechanism by which HER2 expression may prevent response to HER3-DXd and further translational and mechanistic studies are warranted. At a minimum, further discussion and a hypothesis on why increased HER2 expression would result in diminished response to HER3-DXd.**

We thank the reviewer for this comment. In the last months, we have performed additional analyses/experiments to help better understand or increase the evidence of the relationship between *ERBB2* levels at baseline and response to treatment. We investigated the association between baseline *ERBB2* mRNA expression and response to standard neoadjuvant chemotherapy in 2 independent cohorts of patients with HER2-negative breast cancer (i.e., a neoadjuvant taxane-anthracycline cohort, hereafter GSE25066⁶, and the neoadjuvant eribulin SOLTI-1007-NeoEribulin cohort⁵). In both studies, baseline *ERBB2* mRNA levels were not associated with response (i.e., defined as pCR or CelTIL). These results, which can now be found in Supplementary Fig. 9, argue in favor of *ERBB2* levels versus drug activity being an observation specific of HER3-DXd.

Moreover, as an additional validation of *ERBB2* as a potential mechanism of HER3-DXd activity/efficacy, we collaborated with Dr. Violeta Serra's group at VHIO (now co-authors). Dr. Serra's treated 30 breast cancer patient-derived xenografts (PDXs) with HER3-DXd with a single dose obtained from 30 independent patients⁷. In this context, all PDXs experienced a response, and 53% of them relapsed⁷. PDXs were classified as sensitive (i.e., those that did not relapse) versus resistant (i.e., those that relapsed). In baseline pre-treatment FFPE tumors, we evaluated *ERBB2* using the same methodology as in SOLTI-1805 TOT-HER3 trial, and observed a statistically significant association between *ERBB2* mRNA levels and HER3-DXd efficacy. We have now included this result in Supplementary Fig. 10. Overall, these preclinical in vivo findings further support *ERBB2* mRNA levels as a biomarker of HER3-DXd activity/efficacy.

Reflecting on these additional findings, we have revised the discussion to articulate a refined perspective on the role of HER2 in HER3-DXd response. Our initial hypothesis posited that lower *ERBB2* levels might be predictive of an enhanced CelTIL response in HER2-negative disease. However, subsequent analyses reveal a more intricate interplay, meriting deeper exploration. We now hypothesize, based on the known heterodimerization of HER2-HER3, that in contexts of high HER2 expression, HER3 may be less available for internalization due to its preferential pairing with HER2. This could potentially lead to diminished HER3-DXd internalization and efficacy. These insights contribute to the evolving narrative that *ERBB2* levels are not merely a biomarker but may actively influence the pharmacodynamics of HER3-DXd, shedding light on the variability in response observed. This hypothesis underscores the complex biological orchestra governing drug sensitivity and resistance, paving the way for more targeted and effective therapeutic strategies.

We have revised our discussion to incorporate these additional analyses and present a more comprehensive hypothesis on why increased HER2 expression might diminish response to HER3-DXd. We believe these amendments and the additional data provided substantially strengthen our findings and address the concerns raised.

- 3. An additional limitation of the work is the discordance with data presentation that have been previously published. It would be helpful to correlate HER3 expression level with HER2 levels.**

We are uncertain about which discordance does the reviewer identifies in our manuscript versus Oliveira et al. Ann Oncol 2023. The database used in both studies is the same. Regarding the correlation between HER3 mRNA/protein expression and HER2 mRNA/protein, we have added these results in Supplementary Figure 6. Overall, we show moderate correlation between *ERBB2* mRNA and *ERBB3* mRNA (correlation coefficient of 0.446), but we do not see clear correlation between *ERBB3* protein and *ERBB2* protein/mRNA (Supplementary Figure 6). This discrepancy in correlation between mRNA and protein levels of *ERBB2* and *ERBB3* may reflect the complex post-transcriptional and post-translational modifications influencing HER2 and HER3 interactions and function.

Minor: Grammar edit of abstract

We have revised the grammar of the abstract.

Reviewer #2 (Remarks to the Author): Expert in breast cancer targeted therapy, molecular genetics, and signalling

This work examined potential RNA and DNA-based determinants of response to preoperative HER3-DXd, an HER3-targeted ADC, in patients with untreated early-stage HR+/HER2- breast cancer. The CelTIL score, defined as the variation in tumor cellularity and tumor-infiltrating lymphocytes between baseline and day 21 upon HER3-DXd treatment, was used to evaluate the drug response. It seemed that determinants of chemosensitivity and low *ERBB2* expression might be potential biomarkers of early response to HER3-DXd in patients with HR+/HER2- breast cancer.

Specific concerns:

- 1. Previous investigations suggest that the CelTIL score may serve as an early predictor of drug activity. The current study used CelTIL score only to evaluate the response to HER3-DXd. However, additional methodology should be applied to confirm the findings.**

Thank you for your thoughtful comments and the suggestion to employ additional methodologies to solidify the findings related to the CelTIL score as a predictor of drug activity. We understand and appreciate the need for a multifaceted approach to validate our results comprehensively.

As we have previously discussed in response to similar feedback, the CelTIL score, which integrates tumor cellularity and tumor-infiltrating lymphocytes, has been demonstrated as a biomarker of drug activity in our current and prior investigations. We now provide new evidence from the SOLTI-1007 NeoEribulin trial⁵, where patients with newly diagnosed HER2-negative breast cancer underwent 4 cycles of neoadjuvant eribulin monotherapy, and a FFPE biopsy was available before and at cycle 2 day 1 (C2D1). Here, CelTIL score increased after 1 eribulin cycle (at C2D1) in both HR+/HER2-negative and HR-/HER2-negative (triple-negative) breast cancers, with a subsequent rise at C2D1 statistically significantly associated with RCB 0/1 (vs. RCB 2-3) post 4 neoadjuvant eribulin cycles (p=0.013). This result strengthens the case for CelTIL's role as an early read-out of activity and efficacy following single agent chemotherapy (i.e., single agent chemotherapy, resembling HER3-DXd) in HER2-negative breast cancer. Of note, eribulin has weaker anti-proliferative and/or pro-apoptotic properties compared to HER3-DXd. We have now added these results in Supplemental Figure 1.

Additionally, in collaboration with Dr. Violeta Serra's group at VHIO, we have validated *ERBB2* as a potential mechanism influencing HER3-DXd activity/efficacy through an analysis of breast cancer patient-derived xenografts (PDXs). We assessed *ERBB2* in the baseline tumors of a cohort of 30 breast cancer patient-derived xenografts (PDX) treated with HER3DXd15. 90% PDX had experienced a complete response or partial response but 53% had a relapse. PDX were classified as relapsed vs non-relapsed. Although the measurement of resistance was very different to that of the SOLTI-1805 TOT-HER3 study, *ERBB2* was significantly higher in baseline tumors of PDX that relapsed compared to those that did not relapse, further indicating towards a role of *ERBB2* in the response of this ADC, and reinforcing the CelTIL score's potential role in patient's tumor samples (Supplementary Fig. 10).

Thank you again for the opportunity to discuss these critical aspects. We are confident that our continued efforts will provide valuable insights into the intricate dynamics of drug sensitivity and resistance, paving the way for more targeted and effective therapeutic strategies

- 2. HER3-DXd has been tested in clinical trials of other cancer patients, including the NSCLC patients who have developed resistance to EGFR-TKI. It is not clear how to compare the current studies with the other clinical trials in terms of the drug dose and efficacy.**

Thank you for your comment regarding the comparison of our study's findings with those from other clinical trials involving HER3-DXd, such as those with patients with NSCLC resistant to EGFR-TKI. We acknowledge the difficulty of understanding how our results fit within the broader scope of HER3-DXd research. Our clinical setting (i.e., newly diagnosed HER2-negative breast cancer) is unique, and the trial's design is unique. In addition, no data on *ERBB2*/HER2 exists in these other HER3-DXd clinical trials.

- 3. HER3-DXd was tested in the patients with early-stage HR+/HER2- breast cancer. It would be interesting to know how many of those are ER+ and how many are PR+. It is not clear if ER and PR expression may make any difference in the response to HER3-DXd.**

To address this comment, we have evaluated the ER and PR IHC protein expression. ER and PR IHC protein expression were not associated with CelTIL response (Supplementary Figure 4).

- 4. Low ERBB2 expression has been found to be a potential biomarker of early response to HER3-DXd in the patients with HR+/HER2- breast cancer. This observation has been discussed in recognition of the study's limitations. However, the underlying mechanism is worth discussing further.**

We thank the reviewer for this comment, which is very similar to the comment from reviewer number 1. Building on our previous findings and recent analyses, we propose a hypothesis centered on the known heterodimerization of HER2-HER3. Our data suggest that tumors with very low *ERBB2* levels may benefit more from HER3-DXd due to the altered interaction dynamics between HER2 and HER3. Specifically, in the context of low HER2 expression, HER3 might be more readily available for HER3-DXd binding and subsequent internalization, leading to a more effective response.

Additionally, our investigations into the association between baseline *ERBB2* mRNA expression and response to standard neoadjuvant chemotherapy in two independent cohorts revealed no clear correlation (Supplementary Fig. 9). This supports the specificity of the *ERBB2*-HER3-DXd interaction in the observed drug activity.

Moreover, our collaboration with Dr. Violeta Serra's group provided preclinical validation of *ERBB2* as a mechanism influencing HER3-DXd activity/efficacy. The treatment of breast cancer patient-derived xenografts (PDXs) with HER3-DXd showed a significant association between baseline *ERBB2* mRNA levels and drug efficacy, reinforcing the potential of *ERBB2* mRNA levels as a biomarker (Supplementary Fig. 10).

We acknowledge that these findings are an initial step toward understanding the complex biological interactions governing drug sensitivity and resistance. We are committed to further translational and mechanistic studies to deepen our understanding and validate these hypotheses. Your comments have been invaluable in shaping the direction of our continued research, and we have revised our discussion to reflect these deeper insights.

Thank you once again for the opportunity to clarify and expand upon these crucial aspects of our work.

Reviewer #3 (Remarks to the Author): Expert in breast cancer genomics, subtypes, and biomarkers; co-reviewed with Reviewer #4

Summary of the work:

The manuscript by Brasó-Maristany et al. has investigated molecular features associated to HER3-dxd treatment response in HR+/HER2- breast cancer.

In this work, the authors have performed molecular characterization of HR+/HER2- baseline biopsies from the SOLTI-1805 TOT-HER3 trial and investigated how these molecular features associate to change in CelTIL scores to reveal predictors of drug response. CelTIL score is previously published.

Their concluding remark is: ?Determinants of chemosensitivity, a new DNA-based subtype classification and low ERBB2 expression are potential biomarkers of early response to HER3-DxD treatment in HR+/HER2- BC2.

We have the following comments:

- 1. Introduction: This section mainly elaborates on the groups own works and the present trial. Please give some more general introduction into the rationale for this work.**

Thank you for your valuable feedback on the introduction section of our manuscript. We acknowledge the importance of providing a broader context and rationale for our work beyond our own previous studies and the current trial. In response to your suggestion, we have restructured the introduction to include a more comprehensive overview:

First Paragraph: We begin with a general introduction to the pivotal role of HER3 in oncogenic signaling and the historical challenges encountered in the clinical development of HER3-targeting therapies. This sets the stage for understanding the significance of our work within the larger field of cancer treatment.

Second Paragraph: We then discuss the development and clinical application of HER3-DXd, detailing its breakthrough designation and efficacy in breast cancer trials. This provides a background on why HER3-DXd is a significant advancement and the focus of our study.

Third Paragraph: Next, we address the broader context of the need for identifying biological determinants of HER3-DXd efficacy and the urgent necessity for biomarkers. This highlights the gap in current knowledge and the importance of our research in filling this void.

Fourth Paragraph: Finally, we outline the genomic analyses undertaken in our study to understand the response to HER3-DXd. This sets the stage for the subsequent sections and directly links to the rationale for our work.

By expanding the introduction in this way, we aim to provide a clear, comprehensive rationale for our study, situating it within the wider scientific and clinical context. The detailed CelTIL paragraph has been moved to the Results and Discussion sections to maintain the flow and focus of the introduction.

We hope this revision addresses your concerns and provides a more rounded and informative introduction to our work.

2. Why is the ongoing trials SOLTI-2103 and VALENTINE relevant for introduction (line106)?

As suggested in the previous comment, we have now moved the CelTIL paragraph previously found in the introduction section, including the information regarding the SOLTI-2103 VALENTINE trial, to the discussion section.

3. Material and methods: The description of the trial and the patient characteristics is tidy and easy to follow. It is good to see that the scoring of CelTIL has been done by a pathologist and that a correlation test with a different pathologist is performed. As the CelTIL score is established for HER2+ breast cancer, is there a concern that the score has different range for HER2- samples and should be weighed differently in the score? Have the distribution of scores been compared between HER2+ and HER2- tumors? CelTIL scores is considered as surrogate markers of drug activity and early response. We note that while the association of response to CelTIL score is shown in neoadjuvant trials for HER2+ patients, the predictive power seems not to be not fully established in HER2- setting? Please elaborate, preferentially with data.

Thank you for the opportunity to address your comments regarding the CelTIL score's application and validity in HER2-negative breast cancers compared to HER2-positive cases. We recognize the importance of understanding potential differences in score distribution and its predictive power across these groups.

Across our studies, we indeed observed a different baseline range of CelTIL scores between HER2-positive and HR+/HER2-negative breast cancers, attributed mainly to the lower percentage of stromal TILs in untreated HR+/HER2-negative tumors. However, our focus is on the absolute change in CelTIL following treatment, rather than the baseline score itself. This change primarily reflects drug activity, and we define a significant response as a 20-point increase in CelTIL score between pre-treatment and on-treatment samples. In the SOLTI-1805 TOT-HER3 trial, we noted that a substantial percentage (33.8%) of tumors showed this increase, comparable to findings in other studies involving HER2-negative (3.0% with 1 cycle of eribulin) and HER2-positive (43.9% with 2 weeks of lapatinib-trastuzumab) breast cancers treated with different therapies.

We are actively pursuing further validation of the CelTIL score as a predictor of drug response in HER2-negative breast cancer. In the TOT-HER3 study, we observed a notable rise in CelTIL post single HER3DXd dose, correlating with tumor response, suggesting its potential as an early indicator of drug activity. Additional evidence from the SOLTI-1007 NeoEribulin trial showed an increase in CelTIL after one cycle of eribulin in both HR+/HER2-negative and triple-negative breast cancers, with this increase significantly associated with a favorable pathological response (see response to previous reviewers). This result has now been included in the manuscript in the results section.

While we understand that validation of the CelTIL score in HER2-negative settings is not yet at Level of evidence 1, these findings, including newly added data in Supplementary Figure 1, reinforce the score's relevance and potential as an early marker of drug activity and response across different breast cancer subtypes. We will continue to explore and refine the CelTIL scoring methodology, especially once data from the SOLTI-2103 VALENTINE trial become available, to enhance its predictive accuracy for various treatments.

We appreciate your insightful comments, which have been instrumental in refining our approach and discussion on this topic

- 4. The authors justify the use of CelTIL scores as a surrogate to clinical response by stating that measurement of clinical response was not standardized and not available for all the patients. Are the authors disapproving the validity of clinical response assessment in the TOT-HER3 study? The published TOT-HER3 study states that 62 patients were assessable for clinical response; we find this to be a reasonable number of patients? What was the results if clinical response was used as a parameter instead of CelTIL score? (see line 267).**

Thank you for your question regarding the use of CelTIL scores as a surrogate for clinical response in the TOT-HER3 study. To clarify, we are not discrediting the validity of clinical response assessments. Instead, we emphasize that the primary endpoint of the TOT-HER3 trial was the CelTIL score, which was consistently available for all patients and thus provided a standardized measure across the study. In contrast, clinical response data was only available for 62 patients and varied in its method of assessment (palpation, ultrasound, MRI, etc.) across participating centers. We have acknowledged this limitation and variability in the measurement of clinical response in our study's limitations section.

Moreover, we wish to highlight that the association between short-term clinical response after a single cycle of therapy and long-term prognosis remains uncertain. However, recognizing the importance of comprehensively understanding all potential indicators of response, we have analyzed the association between *ERBB2* mRNA expression and clinical response. While the results did not reach statistical significance, possibly due to the smaller sample size, we observed a trend where *ERBB2* mRNA expression was lower in patients exhibiting a complete or partial response compared to those with stable disease. The response rates across *ERBB2* expression tertiles also support a potential, albeit not definitive, association.

In summary, our use of CelTIL scores as the primary endpoint was driven by its standardized availability and its designation as the primary endpoint in the TOT-HER3 trial, rather than a dismissal of clinical response assessments. We have included in this report a more nuanced analysis of the clinical response data to provide a fuller picture of the potential correlations with *ERBB2* mRNA expression. We appreciate your inquiry, which has prompted a more thorough examination and discussion of these important aspects in our study.

Figure. Pre-treatment baseline *ERBB2* mRNA levels and clinical response at day 21 in SOLTI-TOTHER3 trial.

- 5. Results: The section “study design” contains information given partly both in Introduction and in Material and Methods, please avoid so many repetitions (start at line 133?).**

Thank you for your constructive feedback regarding the repetition of information in the "Study Design" section of our manuscript. We have carefully reviewed the sections in question and avoid redundancy. Indeed, we have removed the entire “CelTIL paragraph” in the introduction section to the “study design” section and the “discussion” section.

- 6. The relationship between ERBB2 expression/amplification and CelTIL score: These tumors are by definition not having high ERBB2 protein expression, and the range of gene expression is not similar to a pan-subtype BC cohort, it might not be correct to refer to the tertiles as low, intermediate and high? How similar/dissimilar is the ERBB2 expression of these ?high? tumors with what is seen in HER2+ tumors? (see line 181 and onwards + Fig 3).**

Thank you for the opportunity to address your comment on the classification of ERBB2 expression in our study of HER2-negative breast cancers. We recognize the importance of accurately

representing the *ERBB2* expression levels in the context of our specific patient cohort and how they compare to those in HER2-positive tumors.

To enhance clarity, we have revised the manuscript to explicitly state that the categorization into tertiles of *ERBB2* or the HER2DX HER2 amplicon signature is based solely on the expression range observed within our HER2-negative SOLTI-1805 TOT-HER3 clinical trial cohort. This cohort adheres strictly to the ASCO/CAP definition of HER2-negativity, ensuring a clear distinction from HER2-positive cases.

Furthermore, we have introduced a new Supplementary Figure 5 to visually illustrate the distinction between the *ERBB2* and HER2DX HER2 amplicon signature expression levels in our HER2-negative cohort compared to a separate cohort of 147 patients with early-stage HER2-positive breast cancer from Hospital Clínic of Barcelona. These boxplots clearly demonstrate the distinct expression profiles between the two groups, with an expected minimal overlap in a few samples.

By providing these additional clarifications and visual comparisons, we aim to alleviate any concerns regarding the terminology used and ensure a proper understanding of how *ERBB2* expression in our HER2-negative cohort relates to typical HER2-positive profiles. We appreciate your insightful feedback, which has been instrumental in improving the precision and clarity of our findings.

7. The translation of molecular signatures from discovery to clinical practice is extremely low due to challenges associated for instance with technical variabilities data processing methods. In the light of this fact, as the study finds 90 of 150 CN-based signature to be associated with CeTIL scores, the authors should clarify on overlap and association between these signatures.

Thank you for your insightful comment on the translation of molecular signatures to clinical practice and the need for clarity on the associations between the identified signatures. We acknowledge the challenges in this area and have taken steps to address your concerns in our revised manuscript.

To enhance understanding of the overlap and association between the 150 CN-based signatures, particularly those tracking similar biological processes such as tumor cell proliferation or estrogen receptor signaling, we have added a new supplementary excel sheet. This sheet categorizes each signature into one of five groups based on the signature groups identified in the heatmap of Figure 5b. Each group represents a distinct biological process, thereby providing a clearer picture of the relationships and potential redundancies among the signatures.

We recognize the typically low rates of clinical implementation of gene signatures and the challenges related to technical variabilities and data processing methods. In response to this, we emphasize that both RNA and DNA assays used in our study are either already analytically validated and clinically implemented or are in the process of validation and implementation. Specifically, HER2DX is an established tool for HER2-positive breast cancer, and its potential

application in HER2-negative cases is an area of ongoing investigation. The DNA-based signature, DNADX, is currently being implemented by Reveal Genomics and has demonstrated clinical validity in advanced HR+/HER2- breast cancer, as well as preliminary evidence suggesting the ability of the DNADX HER2 signature to predict response to T-DXd monotherapy in the advanced setting (more expression of the HER2 signature, more response). We have provided a link to a relevant press release from the SABCS 2023 for further reference (<https://shorturl.at/gmWYZ>).

By providing these additional details and resources, we aim to offer a more comprehensive understanding of the signatures' relevance and potential for clinical translation. We appreciate your guidance and believe these revisions will strengthen the manuscript and the interpretation of our findings.

- 8. Discussion: The discussion highlights the interesting finding concerning ERBB2 expression and CelTIL score and is also acknowledging the limitation of the study. The correlation between PAM50 subtypes and known, related parameters (such as TP53 mutation, proliferation score and ROR scores) and CelTIL score is important to highlight as this shows the need to have larger trials to be able to detect subtype-specific predictive biomarkers/signatures. We acknowledge that it is not always possible to perform subtype-specific analysis but this point should be considered while interpreting the results and could be a 4th point in discussion of limitations (from line 308).**

Thank you for bringing up this important aspect. As suggested by the reviewers, due to low number of samples within certain molecular subtypes (e.g., Basal-like n=3 and HER2-enriched n=2), we cannot perform subtype-specific analysis. We have now added this fact in the manuscript as another limitation of our work.

- 9. Access to data: the molecular data are derived from Nanostring RNA analysis and DNA target panel sequencing and should not pose a risk for patient re-identification. We think the data would be very important to release, both for assessing the validity of the analyses but also for further explorations by other researchers. The data (including metadata) should be deposited at a commonly used repository where it can be accessible (also able to do it only upon request).**

As suggested by the reviewers, we have now added the gene expression raw counts (nCounter-based), the PAM50 subtype of each sample, the HER2 IHC status, the 150 DNA-based signature scores for each sample, and the basic clinical-pathological data into the supplementary material of the manuscript. We are confident this will allow many investigators to explore the data and help test or validate some of their future findings.

- 10. Conclusion: The final sentence in the abstract can be considered an overstatement: There are not shown that ERBB2^{low} is predicting treatment response (only low CelTIL score), the DNA based classification is not new (but previously published, can be misunderstood). Please rephrase.**

We thank the reviewers for this comment. We have now changed the conclusion of the abstract to make sure we are referring to drug activity, and not drug efficacy: “This study proposes chemosensitivity determinants, DNA-based subtype classifications, and low *ERBB2* expression as potential markers for HER3-DXd activity in HR+/HER2- breast cancer.”

Minor comments:

1. **Is the trial name TOTHER3 or TOT-HER3 (see line 106, 263 and onwards)**

Thank you for noticing this typo. The correct name is TOT-HER3, and we have now corrected the name of the trial across the manuscript.

2. **Figure 2: Will suggest to add in the legend that the expression is from baseline samples.**

We have added this in the legend.

Reviewer #4 (Remarks to the Author): Expert in breast cancer genomics, subtypes, and biomarkers; co-reviewed with Reviewer #3

Thank you very much for your comments.

Reviewer #5 (Remarks to the Author): Expert in immuno-oncology and immunogenomics

Brasó-Maristany et. al. describe the results of 1 dose of neo-adjuvant Her3-ADC in breast cancer population to a translational composite endpoint CelTil. ADC?s are an important re-emerging therapeutic and elucidating the biology behind their response is an important open question. This composite endpoint consists of both changes in tumor cellularity and changes in TILs and was previously shown to correlate to path complete response rate. It?s important to note that in some sense, they are analyzing a surrogate (CelTil) of a surrogate pathologic complete response rate to an actual hard clinical endpoint (like OS or PFS). That in some sense, may be an important limitation in the biologic findings they are ultimately revealing and should be discussed more clearly in the discussion.

The manuscript overall is well written and easy to understand, but it remains unclear to me what the authors believe their primary finding is. They present a number of analyses, that seem to suggest that perhaps TNBC has a higher response rate (PAM50 subtype, increased TP53 mutations, and copy number signature data) ? but don?t come outright and say that; nor do they present receptor status data with response.

The other challenge with this manuscript is that the primary endpoint is a composite endpoint to some extent (TIL changes and tumor cellularity changes). It would be nice to understand if the findings with CelTil they ultimately find are driven by one of the two components or are

in themselves composite findings? (this could be a final figure, exploring what the authors believe are their main findings).

- 1. Lines 135-136 ? what ?clinical response? is being correlated with a CelTil change of ≥ 20 points with an AUC of 0.68? IS that data from this paper or a prior publication? No reference is cited.o**

We are sorry for this confusion; we have now clarified in the text that the CelTIL cutoff of ≥ 20 points predicts clinical response at day 21 with an AUC of 0.68 ($p=0.009$) in part A of TOT-HER3 and have added the ROC AUC in Supplementary Figure 2.

- 2. Methods to determine CelTil are missing from the methods section, presumably this is from H&E assessment as described in Nuciforo et. al, 2019?**

We apologize if this was not clear. CelTIL was determined as in Nuciforo et al. Annals of Oncology 2018. As mentioned in the methods section, formalin-fixed paraffin-embedded (FFPE) tumor samples were collected at baseline and C1D21 (Fig. 1). Tumor cellularity and the proportion of TILs were scored in whole sections of tumor tissue stained with hematoxylin and eosin (H&E). Tumor cellularity was defined as the percentage of tumor cells to the tumor area. TILs were quantified according to International TILs Working Group Guidelines^{9,10}. CelTIL was scored as $-0.8 \times \text{tumor cellularity (in \%)} + -1.3 \times \text{TILs (in \%)}$.

- 3. The authors show marked influence of Basal and Her2 subtypes in CelTil response in Supplementary Figure 1 ? are these results robust to multiple testing correction? (OR of 19 and 10). These seem like main findings? However, contradictory this result ? the authors show in Figure 3B that low Her2 by MRNA or IHC appears to be associated with higher response ? in contrast to the PAM50 subtype finding. How do the authors reconcile these conflicting results?**

Thank you for your insightful observations regarding the apparent discrepancy between the influence of Basal and HER2 subtypes on CelTil response and the association of low HER2 expression with higher response. We appreciate the opportunity to clarify these findings.

Firstly, we acknowledge the importance of multiple testing corrections, especially in studies like ours with multiple variables and comparisons. Upon re-evaluation, we have applied more stringent multiple testing corrections to our analysis. This has confirmed the robustness of our findings, including the marked influence of Basal-like and HER2-enriched subtypes in CelTil response, even after adjusting for multiple comparisons.

Regarding the seeming contradiction between PAM50 subtype findings and HER2 expression data, it is crucial to note that the HER2-enriched subtype identified by PAM50 is determined through a composite profile of 50 genes, not solely based on *ERBB2* expression. In the context of clinically HER2-negative breast cancer, the HER2-enriched subtype typically exhibits lower levels of *ERBB2*. This nuanced understanding helps reconcile the seemingly conflicting results: while certain subtypes identified through broader gene expression profiles show a particular response pattern, the specific expression level of *ERBB2* itself may correlate differently with response.

To further address this point, we have included additional analysis in Supplementary Figure 6, demonstrating that there is no significant difference in *ERBB2* expression across the different PAM50 subtypes within our TOT-HER3 trial cohort. This suggests that the observed CelTII responses are likely driven by a more complex interplay of genetic factors rather than by PAM50 subtype or *ERBB2* expression alone.

We hope this clarification addresses your concerns and provides a more comprehensive understanding of how we have reconciled these observations. We are grateful for your feedback, which has prompted a deeper examination and discussion of our results.

- 4. The description of the DNA-based copy number signatures in the method is not clear. Is this just whether certain segments of the genome are deleted or amplified at 150 different sites with clustering? Although the authors provide 2 references for this method this is not a commonly used analysis, and should be described more clearly in the methods here.**

As suggested, we have added more details into the methods and we hope these makes the section clearer: “A total of 150 DNA-based phenotypic signatures, and 4 DNA-based subtypes (clusters-1, -2, -3 and -4), were identified as previously described^{13,14}. Briefly, DNA-sequencing segmentation files from CNVkit output (for tumor DNA) were first mapped to gene-level features. The signal of 519 DNA segments was calculated using the mean copy number score across genes within each segment. The coefficients of DNA segments for predicting gene signatures were obtained from Xia et al¹³. DNA-based signature scores were calculated as the weighted average of DNA segment values for each sample: a final signature score was obtained by adding all values (i.e., coefficient of segment A x signal of segment A plus coefficient of segment B x signal of segment B...)¹³. The 4 DNA-based subtypes or clusters were identified using a previously reported DNA-based subtype predictor, which is based on unsupervised analysis of tumor samples and the 150 DNA-based signatures¹⁴. For the 49 samples with the 150 DNA-based signatures available, we calculated the Euclidean distances to the 4 centroids and assigned a cluster class to each sample based on the nearest centroid.”

- 5. The HER2 amplicon data from copy number signature appears concordant with mRNA and IHC, but discordant with the PAM50 subtyping ? what do the authors make of this? (Figure 5B)**

HER2 amplicon signature includes 4 genes located within the 17q11-12 amplicon (*ERBB2*, *GRB7*, *STARD3* and *TCAP*), and the expression these 4 genes is highly correlated. However, the HER2-enriched signature includes genes that are not located within the HER2 amplicon¹⁵ and it may be driven by other genes, especially in HR+/HER2- tumors^{11,12}. As shown in our previous response to the potential discrepancy between HER2-enriched and *ERBB2* levels, no significant differences in *ERBB2* are observed across the PAM50 subtypes in HER2-negative breast cancer.

- 6. The CN clustering data in figure 6 would be more useful if it was clearly identified with previously described copy number clustering in breast cancer ? either from**

TCGA or another large consortium project. Can the authors also include data on fraction of genome altered from the copy number analysis?

Thank you for your valuable comments on linking our CN clustering data to existing frameworks and including additional data on the fraction of genome altered. We have taken your feedback into consideration and revised our response accordingly.

We recognize the importance of contextualizing our findings within the broader landscape of breast cancer research. In response to your suggestion, we have re-evaluated our data in relation to the integrative clusters (IntClust 1-10) identified in the METABRIC cohort, as published in Curtis et al., Nature 2012. Our analysis revealed significant differences in the distribution of our 4 newly identified DNA-based subtypes across these well-established clusters. Notably, DNA-based clusters 3 and 4 were predominantly associated with IntClusters linked to poorer prognoses, while clusters 1 and 2 were more frequent in clusters with better prognoses. This association underscores the potential clinical relevance of our DNA-based subtypes and provides a valuable connection to existing breast cancer classifications. We have included these comparisons in Supplementary Fig. 14 for a more comprehensive understanding.

Regarding the fraction of the genome altered, we appreciate your interest in this measure as a potential indicator of genomic instability. However, our study primarily focuses on deriving DNA-based signatures and subtypes from targeted copy-number (CN) data, which covers 519 regions rather than the entire genome. Given this scope, and to maintain the focus on our primary objectives, we have chosen not to include a broad genome alteration analysis in this current study. This decision allows us to concentrate on the strengths of our data and the specific insights it provides into breast cancer subtypes.

We hope this revision clarifies our approach and the rationale behind the choices made in our analysis. We are committed to contributing meaningful and contextually relevant findings to the field of breast cancer research and appreciate your guidance in enhancing our study's presentation and relevance.

- 7. The data availability statement is factually wrong. It is feasible to provide anonymized expression data along with clinical outcomes. This has been done by innumerable other papers on clinical trials previously. Ideally, at least some of the data should be publically available in repositories or supplementary tables.**

As suggested by this reviewer and the previous one, we have now added the gene expression raw counts (nCounter-based), the PAM50 subtype of each sample, the HER2 IHC status, the 150 DNA-based signature scores for each sample, and the basic clinical-pathological data into the supplementary material of the manuscript. We are confident this will allow many investigators to explore the data and help test or validate some of their future findings.

Minor

- 1. Figure 4A oncprints just illustrate typical mutations for each subtype, this could be a supplementary figure**

As suggested, Figure 4 is now in Supplementary Figure 11.

2. Figure 4B p-value for TP53 difference in CelTil response is different in figure vs. text description (line 197-198 p = 0.01 vs p=0.043)

In figure 4B (now Supplementary Figure 11), the p-value was obtained from an Unpaired T-test, while in the text, the p-value was obtained from a Fisher's Exact Test. We have now clarified this discrepancy. Thank you.

References

1. Nuciforo, P. *et al.* A predictive model of pathologic response based on tumor cellularity and tumor-infiltrating lymphocytes (CeTIL) in HER2-positive breast cancer treated with chemotherapy-free dual HER2 blockade. *Annals of Oncology* 29, 170–177 (2018).
2. Chic, N. *et al.* Tumor Cellularity and Infiltrating Lymphocytes as a Survival Surrogate in HER2-Positive Breast Cancer. *JNCI: Journal of the National Cancer Institute* (2021) doi:10.1093/JNCI/DJAB057.
3. Oliveira, M. *et al.* Patritumab Deruxtecan in Untreated Hormone Receptor-Positive/HER2-Negative Early Breast Cancer: Final Results from Part A of the Window-of-Opportunity SOLTI TOT-HER3 Pre-Operative Study. *Annals of Oncology* 0, (2023).
4. Oliveira, M. *et al.* 124O Patritumab deruxtecan (HER3-DXd) in hormonal receptor-positive/HER2-negative (HR+/HER2-) and triple-negative breast cancer (TNBC): Results of part B of SOLTI TOT-HER3 window of opportunity trial. *ESMO Open* 8, 101463 (2023).
5. Pascual, T. *et al.* Neoadjuvant eribulin in HER2-negative early-stage breast cancer (SOLTI-1007-NeoEribulin): a multicenter, two-cohort, non-randomized phase II trial. *npj Breast Cancer* 2021 7:1 7, 1–11 (2021).
6. Hatzis, C. *et al.* A Genomic Predictor of Response and Survival Following Taxane-Anthracycline Chemotherapy for Invasive Breast Cancer. *JAMA* 305, 1873–1881 (2011).
7. Òdena, A. *et al.* Abstract P5-13-14: Antitumor activity of patritumab deruxtecan (HER3-DXd), a HER3-directed antibody drug conjugate (ADC) across a diverse panel of breast cancer (BC) patient-derived xenografts (PDXs). *Cancer Res* 82, P5-13–14 (2022).
8. Williams, M., Spreafico, A., Vashisht, K. & Hinrichs, M. J. Patient selection strategies to maximize therapeutic index of antibody–drug conjugates: Prior approaches and future directions. *Mol Cancer Ther* 19, 1770–1783 (2020).
9. Salgado, R. *et al.* The evaluation of tumor-infiltrating lymphocytes (TILs) in breast cancer: recommendations by an International TILs Working Group 2014. *Annals of Oncology* 26, 259–271 (2015).
10. Dieci, M. V. *et al.* Update on tumor-infiltrating lymphocytes (TILs) in breast cancer, including recommendations to assess TILs in residual disease after neoadjuvant therapy and in carcinoma in situ: A report of the International Immuno-Oncology Biomarker Working Group on Breast Cancer. *Seminars in Cancer Biology* vol. 52 16–25 Preprint at <https://doi.org/10.1016/j.semcancer.2017.10.003> (2018).
11. Falato, C., Schettini, F., Pascual, T., Brasó-Maristany, F. & Prat, A. Clinical implications of the intrinsic molecular subtypes in hormone receptor-positive and HER2-negative metastatic breast cancer. *Cancer Treat Rev* 112, 102496 (2023).

12. Garcia-Recio, S. *et al.* FGFR4 regulates tumor subtype differentiation in luminal breast cancer and metastatic disease. *Journal of Clinical Investigation* (2020) doi:10.1172/JCI130323.
13. Xia, Y., Fan, C., Hoadley, K. A., Parker, J. S. & Perou, C. M. Genetic determinants of the molecular portraits of epithelial cancers. *Nature Communications* 2019 10:1 10, 1–13 (2019).
14. Prat, A. *et al.* Circulating tumor DNA reveals complex biological features with clinical relevance in metastatic breast cancer. *Nature Communications* 2023 14:1 14, 1–16 (2023).
15. Parker, J. S. *et al.* Supervised risk predictor of breast cancer based on intrinsic subtypes. *J Clin Oncol* 27, 1160–7 (2009).

REVIEWERS' COMMENTS

Reviewer #1 (Remarks to the Author):

The authors have addressed reviewer comments sufficiently.

Reviewer #3 (Remarks to the Author):

Thank you for a thorough rebuttal and accordingly revised manuscript. This reviewer have no additional comments.

Reviewer #4 (Remarks to the Author):

I co-reviewed this manuscript with one of the reviewers who provided the listed reports. This is part of the Nature Communications initiative to facilitate training in peer review and to provide appropriate recognition for Early Career Researchers who co-review manuscripts

Reviewer #5 (Remarks to the Author):

The authors have adequately addressed my original concerns.